# Critical role of interferons in gastrointestinal injury repair

Constance McElrath[1,2], Vanessa Espinosa[3], Jian-Da Lin[2,4,14], Jianya Peng [1,2,15], Raghavendra Sridhar[1,2], Orchi Dutta[2,3], Hsiang-Chi Tseng [2,4], Sergey V. Smirnov[1], Heidi Risman [4], Marvin J. Sandoval[5,16], Viralkumar Davra[1,2], Yun-Juan Chang [6], Brian P. Pollack [7,8,9,10], Raymond B. Birge[1,11,12], Mark Galan [4], Amariliz Rivera [3,13], Joan E. Durbin [4,13] & Sergei V. Kotenko [1,11,12,13 ✉]

The etiology of ulcerative colitis is poorly understood and is likely to involve perturbation of the complex interactions between the mucosal immune system and the commensal bacteria of the gut, with cytokines acting as important cross-regulators. Here we use IFN receptor-deficient mice in a dextran sulfate sodium (DSS) model of acute intestinal injury to study the contributions of type I and III interferons (IFN) to the initiation, progression and resolution of acute colitis. We find that mice lacking both types of IFN receptors exhibit enhanced barrier destruction, extensive loss of goblet cells and diminished proliferation of epithelial cells in the colon following DSS-induced damage. Impaired mucosal healing in double IFN receptor-deficient mice is driven by decreased amphiregulin expression, which IFN signaling can up-regulate in either the epithelial or hematopoietic compartment. Together, these data underscore the pleiotropic functions of IFNs and demonstrate that these critical antiviral cytokines also support epithelial regeneration following acute colonic injury.

[1] Department of Microbiology, Biochemistry, and Molecular Genetics, New Jersey Medical School, Rutgers—The State University of New Jersey, Newark, NJ, USA. [2] School of Graduate Studies, Rutgers—The State University of New Jersey, Newark, NJ, USA. [3] Pediatrics, New Jersey Medical School, Rutgers—The State University of New Jersey, Newark, NJ, USA. [4] Pathology, Immunology and Laboratory Medicine, New Jersey Medical School, Rutgers—The State University of New Jersey, Newark, NJ, USA. [5] Department of Pathology, New York University School of Medicine, New York, NY, USA. [6] Office of Advance Research Computing, Rutgers—The State University of New Jersey, Newark, NJ, USA. [7] Atlanta Veterans Affairs Medical Center, Atlanta, GA, USA. [8] Department of Dermatology, Emory University School of Medicine, Atlanta, GA, USA. [9] Department of Pathology and Laboratory Medicine, Emory University School of Medicine, Atlanta, GA, USA. [10] Winship Cancer Institute, Emory University School of Medicine, Atlanta, GA, USA. [11] Center for Cell Signaling, New Jersey Medical School, Rutgers—The State University of New Jersey, Newark, NJ, USA. [12] Cancer Institute of New Jersey, Rutgers—The State University of New Jersey, Newark, NJ, USA. [13] Center for Immunity and Inflammation, New Jersey Medical School, Rutgers—The State University of New Jersey, Newark, NJ, USA. [14] Present address: Department of Biochemical Science and Technology, National Taiwan University, Taipei, Taiwan. [15] Present address: Department of Medicine, New Jersey Medical School, Rutgers—The State University of New Jersey, Newark, NJ, USA. [16] Present address: Department of Microbiology and Immunology, Weill Cornell Medical College, New York, NY, USA. ✉email: kotenkse@njms.rutgers.edu

Chronic disruption of intestinal homeostasis accompanied by aberrant immune and inflammatory responses underlies the pathophysiology of inflammatory bowel diseases (IBD), such as ulcerative colitis (UC) and Crohn's disease. The etiology of IBD remains elusive and may involve a variety of environmental and genetic factors[1–6]. Heterogeneous manifestations of this spectrum of disorders may reflect the multifactorial nature of IBD and highlight the need for the development of novel treatment options for the subsets of IBD patients refractory to current therapies, which include immunosuppressive drugs and steroids. Although these disorders are described as an "inflammatory" condition, it is unclear how this chronic inflammatory state is triggered and maintained. The current view is based on the assumption that the ongoing inflammation at the mucosal barrier is driven by gastrointestinal (GI) microbes and/or their products. However, it is uncertain whether aberrant inflammatory responses cause the GI barrier disruption or mucosal barrier integrity dysfunction serves as the initiating trigger for IBD. Intestinal epithelial cells (IECs) have drawn the attention as these cells are critical, not only for the maintenance of barrier integrity and antimicrobial defenses but also play an integral part in regulating mucosal immunity[7,8]. Thus, it is also possible that the chronic inflammation observed in IBD results from a defect in epithelial barrier repair rather than from primary immune dysfunction. Moreover, the association of IBD with an increased incidence of colorectal cancer (CRC) also highlights the dysregulated interaction between immune and epithelial cells at this barrier surface and the need to elucidate the underlying etiology of UC. To this end, several mouse models have been generated to study inflammation-induced CRC and colitis. Among these, the robust and highly reproducible azoxymethane/dextran sodium sulfate (AOM/DSS) model has been used to study inflammation-induced CRC and the DSS model to study acute colitis. Studies using these models have identified several cytokines and growth factors important for promoting mucosal healing following DSS-induced injury, including IL-22 and IL-33[9–11].

IFNs represent a family of cytokines consisting of three distinct types: type I (primarily IFN-α/β), type II (IFN-γ), and type III (IFN-λs). Type II IFN is an immune modulator important for resistance to many microbial infections, while type I and type III IFNs are mostly appreciated for their ability to restrict virus infections. These actions are the direct result of their signal transduction cascade, which is initiated upon the binding of each IFN type to its own unique receptor complex. Type I and type III IFNs were initially proposed to be functionally redundant, as they upregulate a similar repertoire of interferon-stimulated genes (ISGs) in IFN-sensitive cells[12]. Furthermore, both are capable of promoting apoptosis and inhibiting cell proliferation[13,14]. Nonetheless, recent research in vivo has begun to reveal distinct, nonoverlapping functions of type I and type III IFNs in both antiviral and antimicrobial defenses[15–18]. For instance, it has been shown that the action of IFNs is compartmentalized in the small intestine of adult mice, with type I IFNs acting on cells restricted to the lamina propria, and type III IFNs acting on epithelial cells[15,17,18]. Moreover, differences in the kinetics and magnitude of ISG induction in response to type I or type III IFNs have also been observed[16,19].

The pleiotropic effects of type I and type III IFNs and their induction by both microbial products and injury-related signals suggest that their expression may be aberrant in IBD and therefore may contribute to the chronic inflammatory state. Several recent studies also point to a potential role for IFNs as important regulators of GI homeostasis[20,21]. Type I IFN signaling during DSS-induced colitis limited the severity of epithelial damage[21]. Type III IFNs have been reported to function in an

anticolitogenic manner[20]. While the importance of type I and type III IFNs to the acute inflammatory response during intestinal inflammation has been investigated, their potential contributions to mucosal healing remain unexplored.

Here, we report that the combined loss of type I and type III IFN signaling enhanced susceptibility to DSS-induced colitis. Unexpectedly, mice lacking both type I and type III IFN receptors, $Ifnar1^{-/-}Ifnlr1^{-/-}$ mice, do not survive even low levels of DSS exposure. This enhanced susceptibility to DSS-induced colonic injury in $Ifnar1^{-/-}Ifnlr1^{-/-}$ mice is associated with extensive destruction of the colonic epithelium, a significant loss of mucin granule-containing goblet cells, progressive weight loss, and enhanced disease activity. Impaired proliferative capacity of IECs in double IFN receptor-deficient mice, resulting in inefficient colonic tissue repair, correlates with DSS-induced mortality. Experiments conducted with bone marrow chimeric mice reveal that IFN signaling in either the epithelial or hematopoietic compartment is sufficient to support epithelial regeneration in the intestinal mucosa following DSS-induced damage and rescue the mice from lethal disease. Moreover, epidermal growth factor receptor (EGFR) signaling, and specifically, IFN-dependent expression of the EGFR ligand amphiregulin (AREG), is identified as a critical pathway for regeneration of the intestinal epithelium and restoration of the mucosal barrier. This study reveals an unanticipated role for IFN signaling in supporting intestinal epithelial regeneration following acute colonic injury. Therefore, compartmentalized and coordinated action of type I and III IFNs within the GI tract is important not only for the effective antiviral protection but also plays a role in the maintenance of mucosal barrier homeostasis.

## Results

**$Ifnar1^{-/-}Ifnlr1^{-/-}$ mice have enhanced susceptibility to DSS-induced colitis.** As IFNs are pluripotent cytokines known to exert tumor-suppressive actions through a variety of mechanisms[22–26], we initially set out to evaluate the role of type I and type III IFNs in colon cancer development in the AOM/DSS model of inflammation-induced CRC. Wild type (WT) and single type I or type III IFN receptor (IFNR)-deficient ($Ifnar1^{-/-}$ or $Ifnlr1^{-/-}$) or double IFNR-deficient ($Ifnar1^{-/-}Ifnlr1^{-/-}$) mice were administered a single intraperitoneal (IP) injection of the carcinogen AOM followed by three cycles of 1.5% DSS in the drinking water for 7 days followed by regular water for 14 days (Supplementary Fig. 1a). Overall, 87% of WT mice and 100% of single IFNR-deficient mice survived to the experimental endpoint (Supplementary Fig. 2a) and had similar tumor burdens in the large intestine (Supplementary Fig. 2b) with a mean tumor burden of 4.6, 4.3, and 2.3 in WT, $Ifnar1^{-/-}$ mice, and $Ifnlr1^{-/-}$ mice, respectively. Only 9% of $Ifnar1^{-/-}Ifnlr1^{-/-}$ mice survived to the experimental endpoint (Supplementary Fig. 2a), indicating an enhanced sensitivity to DSS treatment that required euthanasia of 91% of animals on day 10 or 11 of AOM/DSS treatment (Supplementary Fig. 1a) due to excessive weight loss. Mice that did survive to the experimental endpoint, demonstrated a significantly elevated tumor burden with an average of 9.3 tumors in the large intestine (Supplementary Fig. 2b).

As 1.5% DSS is a relatively mild treatment generally used to induce minor damage to the colonic epithelium[27], the high mortality of $Ifnar1^{-/-}Ifnlr1^{-/-}$ mice following this treatment was unexpected. We therefore chose to use a single 7 day 1.5% DSS treatment (Supplementary Fig. 1b) of single and double IFNR-deficient mice to explore the basis of this markedly enhanced DSS sensitivity in the absence of IFN signaling. As observed in the AOM/DSS model, 88% of $Ifnar1^{-/-}Ifnlr1^{-/-}$ mice again did not survive DSS exposure, with only 12% of these animals surviving to the experimental endpoint,

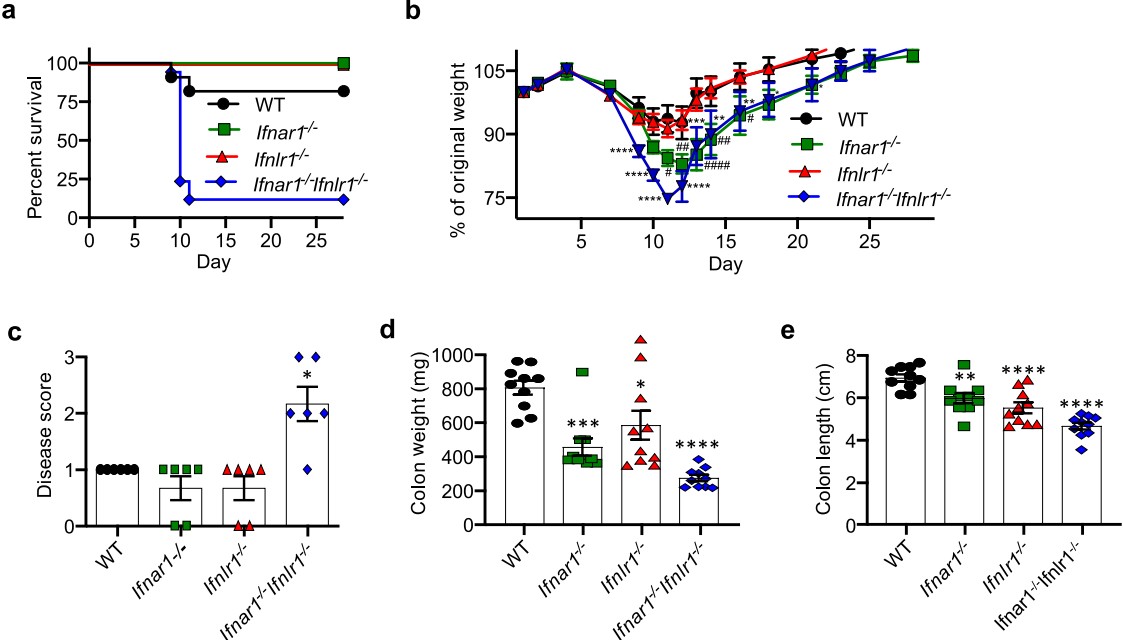

**Fig. 1 Enhanced susceptibility of *Ifnar1⁻/⁻Ifnlr1⁻/⁻* mice to DSS treatment. a–e** 6–8-week-old mice were exposed to 1.5% DSS in drinking water for 7 days, followed by regular drinking water for recovery. Mice were monitored daily for **a** survival and **b** weight loss WT, $n = 11$; *Ifnar1⁻/⁻*, $n = 6$; *Ifnlr1⁻/⁻*, $n = 9$; *Ifnar1⁻/⁻Ifnlr1⁻/⁻*, $n = 17$. In additional sets of experiments, mice were euthanized on day 11 and **c** disease score (WT, $n = 7$; *Ifnar1⁻/⁻*, $n = 6$; *Ifnlr1⁻/⁻*, $n = 6$; *Ifnar1⁻/⁻Ifnlr1⁻/⁻*, $n = 6$), **d** colon weight ($n = 10$), and **e** colon length ($n = 10$) were evaluated. Data are representative of two independent experiments (**a**, **b**) or pooled from two independent experiments (**c–e**). Symbols represent values of individual mice (**c–e**). Quantitative data were analyzed using **b** mixed model or **d**, **e** one-way analysis of variance (ANOVA) or **c** nonparametric Kruskal–Wallis test followed by Bonferroni's (**b**, **d**, **e**) or Dunn's multiple comparisons test (**c**) and represent mean values with SEM. **b** *$P = 0.0217$ (day 18), *$P = 0.0478$ (day 21), **$P = 0.0014$ (day 14), **$P = 0.0096$ (day 16), ***$P = 0.0001$ (day 13), ****$P ≤ 0.0001$ (days 9, 10, 11, 12), an asterisk (*) denotes *Ifnar1⁻/⁻Ifnlr1⁻/⁻*; #$P = 0.0131$ (day 11), #$P = 0.0445$ (day 16), ##$P = 0.0087$ (day 12), ##$P = 0.0043$ (day 14), ####$P ≤ 0.0001$ (day 13), has symbol (#) denotes *Ifnar1⁻/⁻*. **c** *$P = 0.0376$. **d** *$P = 0.0211$, ***$P = 0.0002$, ****$P ≤ 0.0001$. **e** **$P = 0.0082$, ****$P ≤ 0.0001$. $P$ values are for the indicated animal group compared with control group of WT mice (**b–e**).

while 82% of WT mice and 100% of singly IFNR-deficient mice survived after an initial weight loss (Fig. 1a). WT and *Ifnlr1⁻/⁻* mice showed a 5–10% weight loss following the 7 days of DSS treatment, but returned to their original weight by day 20. *Ifnar1⁻/⁻* lost significantly more weight than WT or *Ifnlr1⁻/⁻* animals, but all survived and returned to their starting weight by the end of the 30-day experimental period (Fig. 1b). The 12% of *Ifnar1⁻/⁻Ifnlr1⁻/⁻* mice that survived DSS treatment regained their weight by the end of the experimental period (Fig. 1b).

To evaluate colitis severity after the single cycle of DSS treatment, reduction in colon weight, colon length, and disease score were evaluated on day 11, as this was the time point the majority of *Ifnar1⁻/⁻Ifnlr1⁻/⁻* mice were euthanized due to excessive weight loss. The clinical disease score on day 11 was significantly elevated for *Ifnar1⁻/⁻Ifnlr1⁻/⁻* mice (Fig. 1c), and was marked by severe diarrhea containing visible blood, with only mild disease in cohorts of WT, *Ifnar1⁻/⁻* and *Ifnlr1⁻/⁻* animals. All IFNR-deficient mice demonstrated significant reduction in colon weight (Fig. 1d) and length (Fig. 1e and Supplementary Fig. 3) compared to WT mice, with the reduction in colon weight and length of *Ifnar1⁻/⁻Ifnlr1⁻/⁻* mice being the most pronounced. These experiments showed increased susceptibility to DSS treatment in all IFNR-deficient mice, with *Ifnar1⁻/⁻Ifnlr1⁻/⁻* mice demonstrating the most dramatic phenotype.

**Ifnar1⁻/⁻Ifnlr1⁻/⁻ mice have significant reduction of goblet cells with mucin granules following DSS treatment.** *Ifnar1⁻/⁻Ifnlr1⁻/⁻* mice did not exhibit any abnormalities in colon structure at steady state as judged by evaluating haematoxylin and eosin (H&E) staining (Supplementary Fig. 4a), and periodic acid-Schiff (PAS)/Alcian blue stained sections (Supplementary Fig. 4b) for

tissue morphology and mucus production. PAS/Alcian blue staining displays heavily glycosylated proteins such as mucins revealing mucus layers and colon epithelial cells with mucin-containing granules indicative of mucus-secreting goblet cells. Although, both *Ifnlr1⁻/⁻* and *Ifnar1⁻/⁻Ifnlr1⁻/⁻* mice had thinner mucus layers under homeostatic conditions (Supplementary Fig. 4c, d), WT and all IFNR-deficient mice displayed similar colonic histology, regarding inflammation, extent, regeneration, crypt damage, percent involvement, percent lost epithelium, and percent lost mucin granules during the injury phase (day 5, Supplementary Fig. 5a) and immediately after withdrawal of DSS (day 8, Supplementary Fig. 5b). During the recovery phase, the histology was similar between WT and single IFNR-deficient mice (day 11, Fig. 2a); however, *Ifnar1⁻/⁻Ifnlr1⁻/⁻* mice exhibited significantly more extensive mucosal ulceration (Fig. 2a) with strongly pronounced loss of goblet cells stained positive for mucin granules (Fig. 2b, c), a cell type that plays a key role in supporting epithelial repair[28]. The reduced staining for mucin granules could be due to goblet cell depletion, aberrant degranulation, or impaired regeneration of goblet cells.

It has been reported that commensal microbiota are capable of acting in a colitogenic or protective manner[29–31]. However, cohousing of WT and *Ifnar1⁻/⁻Ifnlr1⁻/⁻* mice did not influence the survival of either strain (Supplementary Fig. 6), indicating that the enhanced pathology in *Ifnar1⁻/⁻Ifnlr1⁻/⁻* mice is not driven by differences in the commensal microbiota.

Taken together, these data demonstrate that the combined loss of type I and type III IFN signaling strongly enhances the susceptibility to experimental colitis, resulting in diffuse destruction of colonic epithelium, reduced number of goblet cells with mucin granules, profound weight loss, and increased mortality.

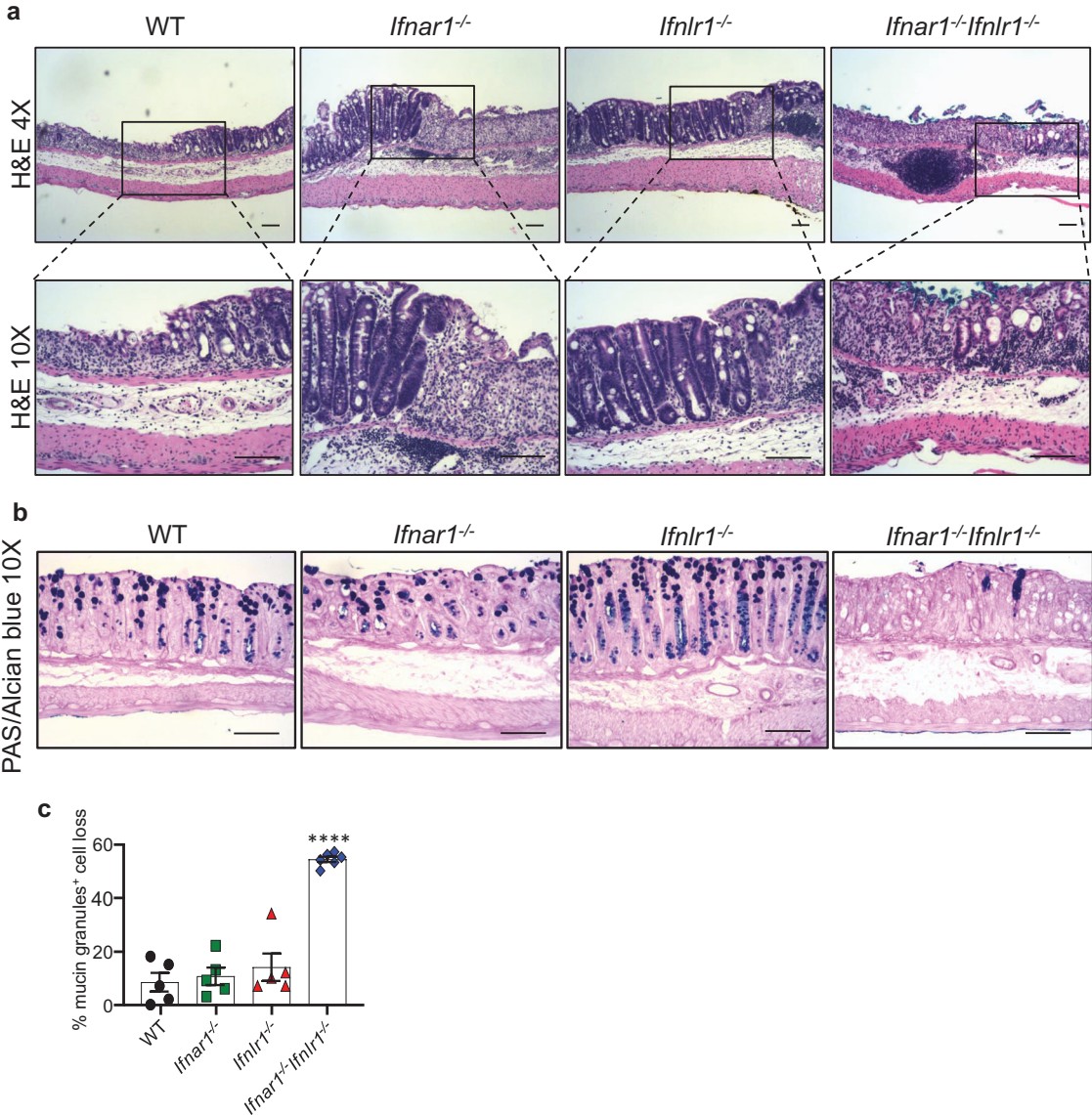

**Fig. 2 Reduced number of goblet cells with mucin granules in the colon of *Ifnar1*⁻/⁻*Ifnlr1*⁻/⁻ mice following DSS treatment. a–c** 6–8-week-old WT, single or double IFNR-deficient mice were exposed to 1.5% DSS in drinking water for 7 days, followed by regular drinking water for recovery, and euthanized on day 11. Representative **a** H&E and **b** PAS/Alcian blue staining for mucin granules-containing goblet cells with **c** quantitated loss of goblet cells stained positive for mucin granules (WT, $n = 5$; *Ifnar1*⁻/⁻, $n = 5$; *Ifnlr1*⁻/⁻, $n = 5$; *Ifnar1*⁻/⁻*Ifnlr1*⁻/⁻, $n = 6$) are shown. Scale bars are 100 µM. Data are representative of three independent experiments. Symbols represent values of individual mice (**c**). Quantitative data were analyzed using one-way ANOVA followed by Bonferroni's multiple comparisons test and represent mean values with SEM. ****$P \leq 0.0001$ and is for the indicated animal group compared with control group of WT mice.

**IFNR-deficient mice have an aberrant inflammatory response following exposure to DSS.** To evaluate the potential effect of IFNR signaling on the acute inflammatory response, we analyzed innate immune cells infiltrating the colon on day 8 of DSS treatment, a time point when all mouse strains showed comparable epithelial damage (Supplementary Fig. 5b). Using the gating strategy shown in Supplementary Fig. 7, we enumerated the neutrophils, monocytes, eosinophils, and CD169⁺ macrophages present in the colon. *Ifnar1*⁻/⁻ and *Ifnlr1*⁻/⁻ mice had significantly more colon-infiltrating neutrophils, monocytes, eosinophils, and CD169⁺ macrophages compared with WT mice (Fig. 3a). Unexpectedly, the innate immune cell infiltration profile of *Ifnar1*⁻/⁻*Ifnlr1*⁻/⁻ mice looked more similar to that of WT mice, but with significantly increased numbers of neutrophils and CD169⁺ macrophages (Fig. 3a). CD169⁺ macrophages were of interest as their infiltration correlates with colitis severity[32,33]. While more of these cells were

present in the colons of all IFNR⁻/⁻ mice, this increase was most pronounced in the *Ifnlr1*⁻/⁻ animals.

While neutrophils play an important role in the elimination of invading microbes, the release of their granules can result in severe pathology, highlighting the need for their function to be tightly regulated[34,35]. We therefore evaluated neutrophil degranulation by measuring in vivo myeloperoxidase (MPO) activity on day 8. MPO activity normalized to the number of colon-infiltrating neutrophils demonstrated that all IFNR-deficient mice had a significant reduction in neutrophil activation (Fig. 3b). In addition, analysis of fecal lipocalin 2 (LCN-2), a biomarker for intestinal inflammation[36] produced by both epithelial cells and infiltrating neutrophils, suggested that all mouse strains developed similar levels of inflammation and with similar kinetics (Fig. 3c). As noted above, the lack of type III IFN signaling alone resulted in the most noticeable increase in immune cell

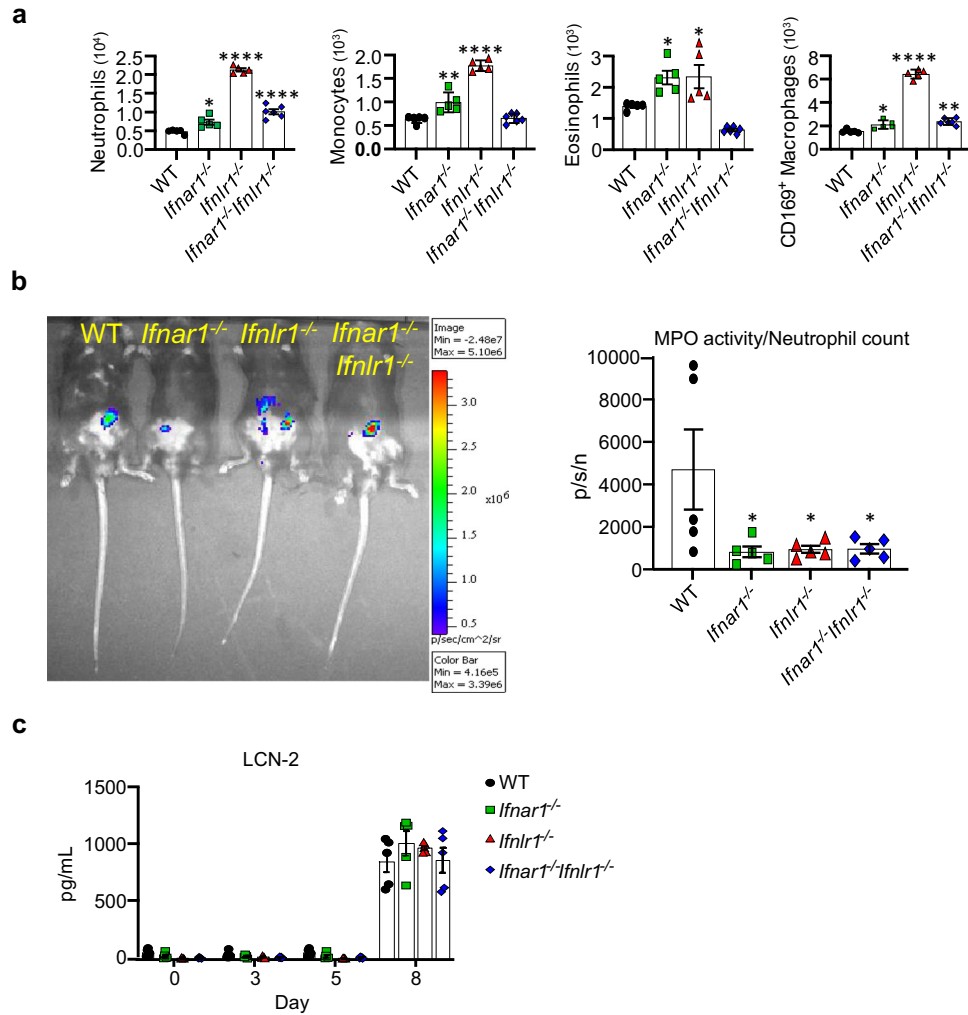

**Fig. 3 Aberrant inflammatory response in the colon of IFNR-deficient mice following DSS treatment. a–c** 6–8-week-old WT, single or double IFNR-deficient mice were exposed to 1.5% DSS in drinking water for 7 days, followed by regular drinking water for recovery. **a** Mice were euthanized on day 8 and immune cell infiltrates in the colon were enumerated by flow cytometry (WT, $n = 5$; $Ifnar1^{-/-}$, $n = 5$; $Ifnlr1^{-/-}$, $n = 5$; $Ifnar1^{-/-}Ifnlr1^{-/-}$, $n = 6$). **b** Neutrophil function was analyzed by evaluating myeloperoxidase (MPO) activity by measuring in vivo luminescence in mice injected with luminescent MPO substrate. Representative images display MPO activity (photon radiance (photons $s^{-1} cm^{-2}$ steradian$^{-1}$; p/s/cm²/sr) presented as pseudo-color images ranging from red (most intense) to blue (least intense) superimposed over gray photographs. Relative MPO activity measured as photon flux (p/s) normalized to the number (p/s/n) of colon-infiltrating neutrophils (live, CD45$^+$CD11b$^+$Ly6G$^+$) is shown ($n = 5$). **c** Fecal lipocalin 2 (LCN-2) levels were measures by ELISA on selected days of experimental DSS-induced colitis ($n = 5$). Data are representative of two independent experiments (**a**, **c**). Symbols represent values of individual mice (**a–c**). Quantitative data were analyzed using **a**, **b** one-way ANOVA or **c** two-way ANOVA followed by Bonferroni's multiple comparisons test (**a–c**) and represent mean values with SEM. **a** *$P = 0.0207$ (neutrophils, $Ifnar1^{-/-}$); **$P = 0.0016$ (monocytes, $Ifnar1^{-/-}$); *$P = 0.0216$ (eosinophils, $Ifnar1^{-/-}$), *$P = 0.0172$ (eosinophils, $Ifnlr1^{-/-}$); *$P = 0.0296$ (CD169$^+$ macrophages, $Ifnar1^{-/-}$), **$P = 0.0011$ (CD169$^+$ macrophages, $Ifnar1^{-/-}Ifnlr1^{-/-}$). **b** *$P = 0.0346$ ($Ifnar1^{-/-}$), *$P = 0.0417$ ($Ifnlr1^{-/-}$), *$P = 0.043$ ($Ifnar1^{-/-}Ifnlr1^{-/-}$). ****$P \leq 0.0001$. $P$ values are for the indicated animal group compared with control group of WT mice (**a**, **b**).

recruitment to the injured colon. However, the presence of an aberrant inflammatory response in single IFNR-deficient mice had no impact on survival, strongly suggesting that the enhanced sensitivity to DSS exposure observed in $Ifnar1^{-/-}Ifnlr1^{-/-}$ mice is not driven by an aberrant inflammatory response.

**Colonic epithelium of $Ifnar1^{-/-}Ifnlr1^{-/-}$ mice demonstrates decreased proliferation following DSS treatment.** Since IECs are an integral part of the intestinal barrier[37], we assessed whether the failure of $Ifnar1^{-/-}Ifnlr1^{-/-}$ mice to recover from DSS exposure (Fig. 1a, b) could be due to a deficiency in mucosal regeneration. Analysis of cell proliferation in the colon of IFNR-deficient mice at steady state, as assessed by Ki-67 immunostaining, revealed no significant difference from WT mice (Fig. 4a). We then evaluated

Ki-67 staining following the injury phase (day 8), when the histological scores were similar between all mouse strains (Supplementary Fig. 5b), and most of the crypts in the large intestine remained intact. WT and $Ifnlr1^{-/-}$ mice averaged 23 and 24 Ki-67$^+$ cells per crypt, respectively, while $Ifnar1^{-/-}$ mice averaged only 15 Ki-67$^+$ cells per crypt, and the mean in $Ifnar1^{-/-}Ifnlr1^{-/-}$ mice was reduced to 8 cells per crypt (Fig. 4b). In addition, the crypts appeared more elongated in WT and $Ifnlr1^{-/-}$ mice then in $Ifnar1^{-/-}$ or $Ifnar1^{-/-}Ifnlr1^{-/-}$ mice, indicating faster cell proliferation (Fig. 4b). This pattern of colonic Ki-67 staining correlates with weight loss (Fig. 1b) since both $Ifnar1^{-/-}$ and $Ifnar1^{-/-}Ifnlr1^{-/-}$ mice show significantly more weight loss following DSS exposure, suggesting that $Ifnar1^{-/-}$ mice also have increased sensitivity to DSS exposure. Nevertheless, while $Ifnar1^{-/-}$ mice have reduced

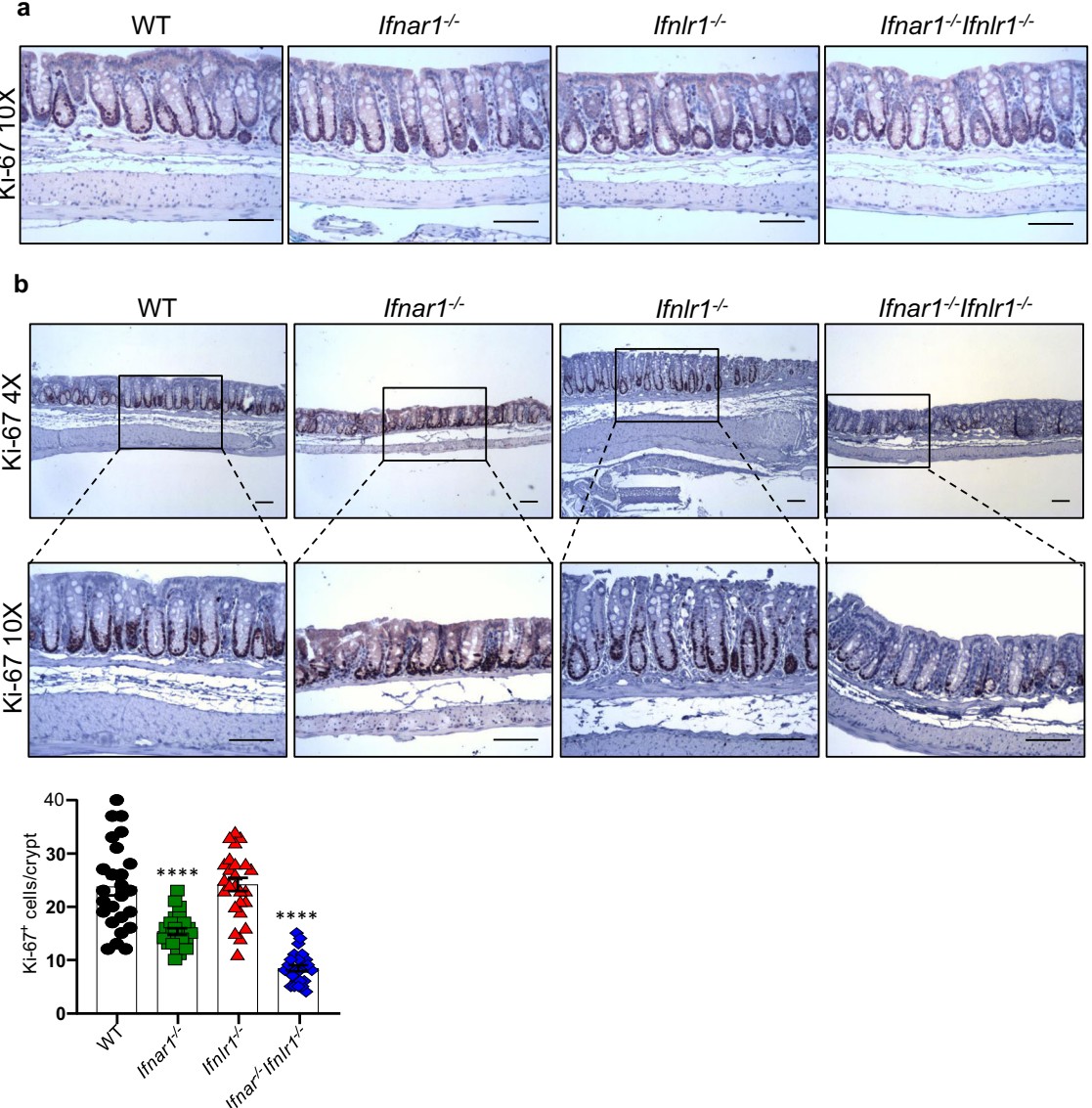

**Fig. 4 Impaired proliferation of colon epithelium in *Ifnar1*$^{-/-}$*Ifnlr1*$^{-/-}$ mice following DSS treatment. a** 6–8-week-old naive WT, single or double IFNR-deficient mice receiving regular water were euthanized and proliferation of colon epithelial cells were evaluated by immunohistochemistry (IHC) staining for Ki-67 ($n = 5$). **b** 6–8-week-old WT, single or double IFNR-deficient mice were exposed to 1.5% DSS in drinking water for 7 days, followed by regular drinking water for recovery. Mice were euthanized on day 8 and proliferation of colon epithelial cells was evaluated by IHC staining for Ki-67 ($n = 5$). Representative colon images and quantified data are shown. Scale bars are 100 μM. Data are representative of two independent experiments. Symbols represent the number of Ki-67$^+$ cells per crypt (five crypt/slide/animal; $n = 5$). Quantitative data were analyzed using one-way ANOVA followed by Bonferroni's multiple comparisons test and represent mean values with SEM. ****$P \leq 0.0001$ and are for the indicated animal group compared with control group of WT mice.

numbers of Ki-67$^+$ cells per crypt, this relative deficiency is not associated with the near 90% mortality observed in *Ifnar1*$^{-/-}$*Ifnlr1*$^{-/-}$ mice. Taken together, these data suggest that the combined loss of type I and type III IFN signaling results in decreased proliferation of colonic epithelial cells following DSS insult, and is responsible for the near total mortality observed in *Ifnar1*$^{-/-}$*Ifnlr1*$^{-/-}$ mice.

**Compartmentalized IFN signaling supports epithelial proliferation following DSS-mediated injury.** Next, we generated a series of bone marrow chimera mice and subjected them to DSS treatment (Supplementary Fig. 1b) to determine whether the reduced regenerative capacity of the colonic mucosa was driven by a deficiency of IFN signaling in the hematopoietic

compartment, epithelial compartment, or both compartments. Flow cytometry analyses of PBMCs obtained from the chimeric mice showed a high percent donor cell engraftment: *Ifnar1*$^{-/-}$*Ifnlr1*$^{-/-}$ mice were reconstituted with donor WT congenic bone marrow CD45.1$^+$ cells (WT→*Ifnar1*$^{-/-}$*Ifnlr1*$^{-/-}$ irradiated hosts) to 94%, and WT congenic mice were reconstituted to 89% with CD45.2$^+$ *Ifnar1*$^{-/-}$*Ifnlr1*$^{-/-}$ bone marrow (*Ifnar1*$^{-/-}$*Ifnlr1*$^{-/-}$→WT irradiated hosts) (Supplementary Fig. 8). Control age-matched WT mice and WT mice reconstituted with WT bone marrow (WT→WT irradiated hosts) did not develop more severe disease following DSS treatment as measured by survival (Fig. 5a) or weight loss (Fig. 5b), and there was no reduction in the extent of colon epithelial cell proliferation (Fig. 5c). Control age-matched *Ifnar1*$^{-/-}$*Ifnlr1*$^{-/-}$ mice and *Ifnar1*$^{-/-}$*Ifnlr1*$^{-/-}$

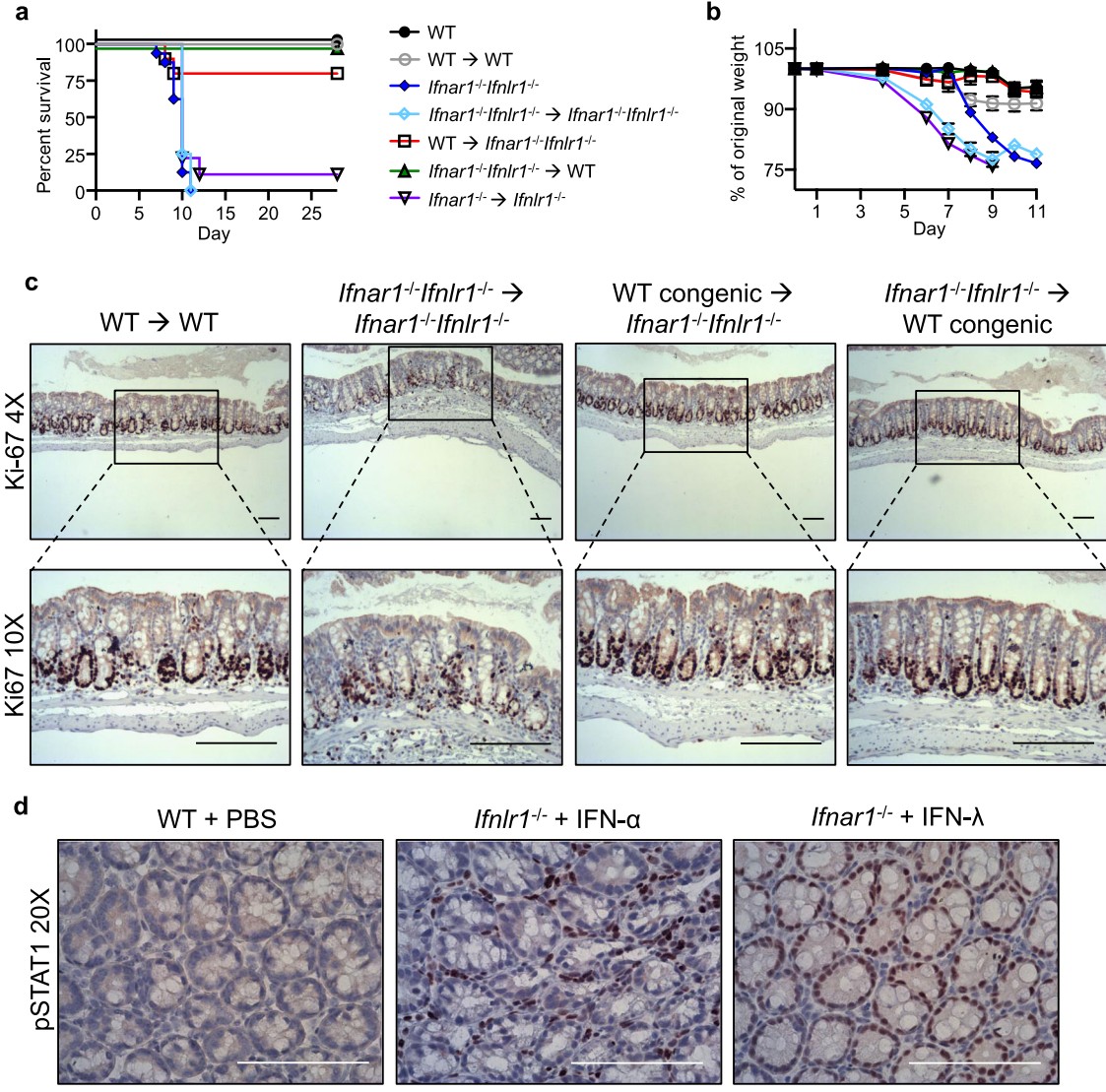

**Fig. 5 Compartmentalized IFN signaling supports epithelial proliferation in experimental DSS-induced colitis. a–c** Indicated bone marrow chimera mice and age-matched WT and *Ifnar1$^{-/-}$Ifnlr1$^{-/-}$* mice were exposed to 1.5 % DSS in drinking water for 7 days, followed by regular water for recovery. Mice were monitored daily for **a** survival and **b** weight loss (WT, $n = 8$; WT→WT, $n = 5$; *Ifnar1$^{-/-}$Ifnlr1$^{-/-}$*, $n = 14$; *Ifnar1$^{-/-}$Ifnlr1$^{-/-}$ → Ifnar1$^{-/-}$Ifnlr1$^{-/-}$*, $n = 4$, *Ifnar1$^{-/-}$Ifnlr1$^{-/-}$*→WT, $n = 10$; WT→*Ifnar1$^{-/-}$Ifnlr1$^{-/-}$*, $n = 10$; *Ifnar1$^{-/-}$*→*Ifnlr1$^{-/-}$*, $n = 9$). Data are presented as mean values with SEM. **c** Mice were euthanized on day 8 and colon epithelial cell proliferation was evaluated by IHC staining for Ki-67 ($n = 5$). **d** 6–8-week-old WT mice were treated with IFN (1.0 μg) or PBS as a control by subcutaneous injection, and euthanized 30 min later. IFN-induced activation of STAT1 was evaluated by IHC staining with antibodies against phosphorylated STAT1 ($n = 5$). Scale bars are 100 μM. Data are representative of two independent experiments (**a–d**).

mice reconstituted with bone marrow from *Ifnar1$^{-/-}$Ifnlr1$^{-/-}$* mice (*Ifnar1$^{-/-}$Ifnlr1$^{-/-}$*→*Ifnar1$^{-/-}$Ifnlr1$^{-/-}$* chimeras) had a dramatic reduction in survival (Fig. 5a), with no DSS-treated mice surviving to the experimental endpoint. Mortality correlated with extensive weight loss (Fig. 5b) and a decrease in the number of Ki-67$^+$ colonic epithelial cells at day 8 of DSS-induced injury (Fig. 5c). These data recapitulate results obtained with cohorts of younger mice and confirm the enhanced sensitivity to DSS treatment (Fig. 1a, b) and impaired recovery in double IFNR-deficient mice regardless of age. However, adoptive transfer of WT congenic bone marrow into *Ifnar1$^{-/-}$Ifnlr1$^{-/-}$* mice (WT→*Ifnar1$^{-/-}$Ifnlr1$^{-/-}$* irradiated hosts) resulted in the survival of 80% of recipients (Fig. 5a) and minimal weight loss (Fig. 5b) following DSS exposure. Notably, adoptive transfer of WT bone marrow into *Ifnar1$^{-/-}$Ifnlr1$^{-/-}$* mice also successfully restored epithelial proliferation (Fig. 5c). Interestingly, 100% of WT congenic mice reconstituted with *Ifnar1$^{-/-}$Ifnlr1$^{-/-}$* bone

marrow (*Ifnar1$^{-/-}$Ifnlr1$^{-/-}$*→WT irradiated hosts) also survived (Fig. 5a) with only mild weight loss (Fig. 5b) and no change in epithelial proliferation (Fig. 5c).

It has been previously demonstrated that, in adult mice, type I and type III IFNs act in a compartmentalized manner within the small intestine; type I IFN induces phosphorylation of STAT1 in CD45$^+$ cells within the lamina propria, while type III IFN induces STAT1 phosphorylation only in epithelial cells[15,18,38]. To confirm that this compartmentalized response to IFNs is conserved in the large intestine, we treated WT mice with IFN-α, IFN-λ or PBS for 30 min, and evaluated patterns of STAT1 phosphorylation and nuclear translocation in colon sections by IHC. As previously reported for the small intestine, treatment with type I IFN triggered STAT1 phosphorylation and nuclear translocation in immune cells within the lamina propria, while treatment with type III IFN resulted in nuclear translocation of phosphorylated STAT1 only in cells lining the colon epithelium (Fig. 5d).

Therefore, data obtained from studies with the bone marrow chimeras, combined with the observed compartmentalized response to IFNs in the colon, suggest that type I IFN signaling in the hematopoietic compartment, or type III IFN signaling in the colonic epithelium is sufficient to support recovery following DSS-induced epithelial damage. This was further established by the finding that adoptive transfer of *Ifnar1*[−/−] bone marrow into *Ifnlr1*[−/−] recipients (*Ifnar1*[−/−]→*Ifnlr1*[−/−] irradiated hosts) recapitulated the observed mortality of *Ifnar1*[−/−]*Ifnlr1*[−/−] mice following DSS treatment (Fig. 5a, b). Colonic epithelial cells in these chimeric animals were deficient for type III IFN receptors, while BM-derived lamina propria cells lacked type I IFN receptors. Only 11% of these chimeric mice survived DSS treatment (Fig. 5a) and all animals suffered dramatic weight loss (Fig. 5b). Taken together, these data highlight the compartmentalized effect of IFN signaling, and demonstrate that type I IFN signaling in the hematopoietic compartment, or type III IFN signaling in the epithelial compartment can mitigate severe DSS-induced injury.

**IFN-regulated AREG expression mediates repair of colonic epithelium following DSS treatment.** The results obtained with bone marrow chimera mice (WT→*Ifnar1*[−/−]*Ifnlr1*[−/−] chimeras) indicated that IFN signaling in the hematopoietic compartment was sufficient to restore proliferation of epithelial cells following injury by DSS, suggesting that this effect is indirect and is likely to be mediated by either type I or type III IFN. Our attention was initially drawn to IL-22, which is important for maintaining proper GI homeostasis[39,40]. A subset of IL-22-producing neutrophils has been shown to contribute to antimicrobial defense and restoration of the mucosal epithelium following DSS treatment[41]. In addition, because we observed aberrant neutrophil function with regard to MPO activity (Fig. 3b), we wanted to determine if double IFNR-deficient mice displayed abnormal IL-22 expression. IL-22 protein levels were measured in colon homogenates at several time points in the DSS-induced colitis model and were found to be similar in all mouse strains with *Ifnar1*[−/−]*Ifnlr1*[−/−] mice showing elevated levels of IL-22 at day 11 (Supplementary Fig. 9a). IL-6 produced by myeloid cells was also reported to promote epithelial cell proliferation[42]. However, levels of IL-6 in colon homogenates did not correlate with the increased sensitivity of *Ifnar1*[−/−]*Ifnlr1*[−/−] mice to DSS (Supplementary Fig. 9b). Thus, impaired tissue repair was unlikely to be due to reduced IL-6 or IL-22 expression in *Ifnar1*[−/−]*Ifnlr1*[−/−] mice, suggesting that another factor regulated by IFN signaling could support recovery following DSS-induced injury.

Since IFN signaling seems to be critical for the colonic recovery from the DSS-induced damage, we used Mx2-Luciferase (Mx2-Luc) reporter mice to evaluate the timing of IFN signaling during acute DSS-induced colitis. These mice, which have intact type I and type III IFN receptors and express the luciferase reporter gene under transcriptional control of the IFN-inducible Mx2 promoter[43], were subjected to the DSS treatment protocol (Supplementary Fig. 1b), and Mx2 promoter-driven luciferase expression was measured by an in vivo imaging system (IVIS). The level of luciferase expression was increased on day 8, and remained elevated through day 22 (Fig. 6a, b), demonstrating that activation of the IFN pathway coincided with recovery following DSS insult. We confirmed induction of IFN signaling in the colon by immunostaining for pSTAT1 in WT mice treated with DSS (Supplementary Fig. 10).

To identify the mechanism by which IFNs were promoting mucosal recovery, we analyzed RNA-seq data obtained from fluorescence-activated cell (FAC)-sorted IECs isolated from 8-day-old pups that were treated with either IFN-α or IFN-λ for 3 days.

Of note, IECs in suckling pups are sensitive to both types of IFNs unlike IECs in adult mice[15]. Accordingly, treatment with either type I or type III IFNs resulted in the transcriptional upregulation of overlapping sets of genes, most of which represented classical ISGs (Supplementary Fig. 11a). Among the 110 genes upregulated more than fivefold by treatment with either IFN (Supplementary Fig. 11b) was AREG. AREG is not encoded by a known ISG, but has been identified as a ligand for the EGFR. Of note, transcription of the gene encoding another EGFR ligand, epiregulin (EREG), was also induced by IFN-λ and to a lower extent by IFN-α (~2-fold). However, based on reads per kilobase of transcript per million mapped reads values, the levels of AREG gene expression are much higher (~60-fold; Supplementary Data 1) than those of the EREG gene at steady state, indicating that changes in AREG expression are likely to have more profound effects in GI homeostasis. Signaling through the EGFR pathway is known to be important for regulating cell proliferation, survival, and motility, and the significance of AREG function at mucosal barrier surfaces is just beginning to emerge[44]. We therefore evaluated AREG expression on day 11 of the DSS treatment protocol using RNA in situ hybridization (RNAscope). The upregulation of AREG transcripts was observed in the epithelium, localized around ulcerated regions of the large intestine in WT but not *Ifnar1*[−/−]*Ifnlr1*[−/−] mice (Fig. 6c). This differential expression of AREG transcripts was confirmed by qPCR in colon homogenates (Fig. 6d). These results revealed impaired AREG upregulation in the large intestine of *Ifnar1*[−/−]*Ifnlr1*[−/−] mice following DSS treatment, despite the presence of similar levels of AREG transcripts in colons of WT and *Ifnar1*[−/−]*Ifnlr1*[−/−] mice under homeostatic conditions (Supplementary Fig. 12a, b).

Epithelial cells have been shown to constitutively express AREG[45], so we next asked whether IFN treatment can augment AREG expression in epithelial cell lines. Treatment of murine IECs (mIECs)[46] with mIFN-λ2, and the combined treatment of mIECs with mIFN-α2/mIFN-λ2 significantly upregulated transcription of AREG (Fig. 6e) as well as ISG15 (Supplementary Fig. 13a), a classical ISG included as a positive control. This IFN activity was not limited to IECs, as murine lung epithelial cells (MLE-15) also upregulated AREG transcription following IFN treatment (Supplementary Fig. 13b; effects of IFNs on classical ISG OAS2 transcription were included as a positive control). The importance of AREG production by hematopoietic cells following mucosal injury has previously been demonstrated[10,47,48], however regulation of AREG expression by IFNs in cells of hematopoietic origin has not been explored. We found that treatment with mIFN-α2 and the combined treatment with mIFN-α2/mIFN-λ2 significantly upregulated AREG transcript expression in FAC-sorted CD45[+] bone marrow-derived cells (Fig. 6f) in addition to classical ISGs (Supplementary Fig. 13c). These data demonstrate that type I and type III IFNs can upregulate AREG expression in both epithelial and hematopoietic cells.

Type III IFNs have been shown to promote cell motility[49–52]. We investigated whether this activity might be mediated through EGFR signaling. For these experiments, we used immortalized mIECs that were shown to recapitulate several characteristics of natural epithelium including their ability to polarize, form tight junctions and microvilli, and respond to IFN[46]. Type III IFNs were also reported to inhibit proliferation of CRC-derived cell lines[53]. We observed no effects of IFN-λ or AREG on proliferation of mIECs within 48 h of the treatment (Supplementary Fig. 14a). However, both IFN-λ and AREG accelerated gap closure in a scratch assay (Supplementary Fig. 14b) and promoted migration of mIECs through transwells (Supplementary Fig. 14c). The stimulation of cell motility by either IFN-λ or AREG was blocked by EGFR kinase inhibitor gefitinib (Supplementary Fig. 14b). These data, combined with fast upregulation of AREG

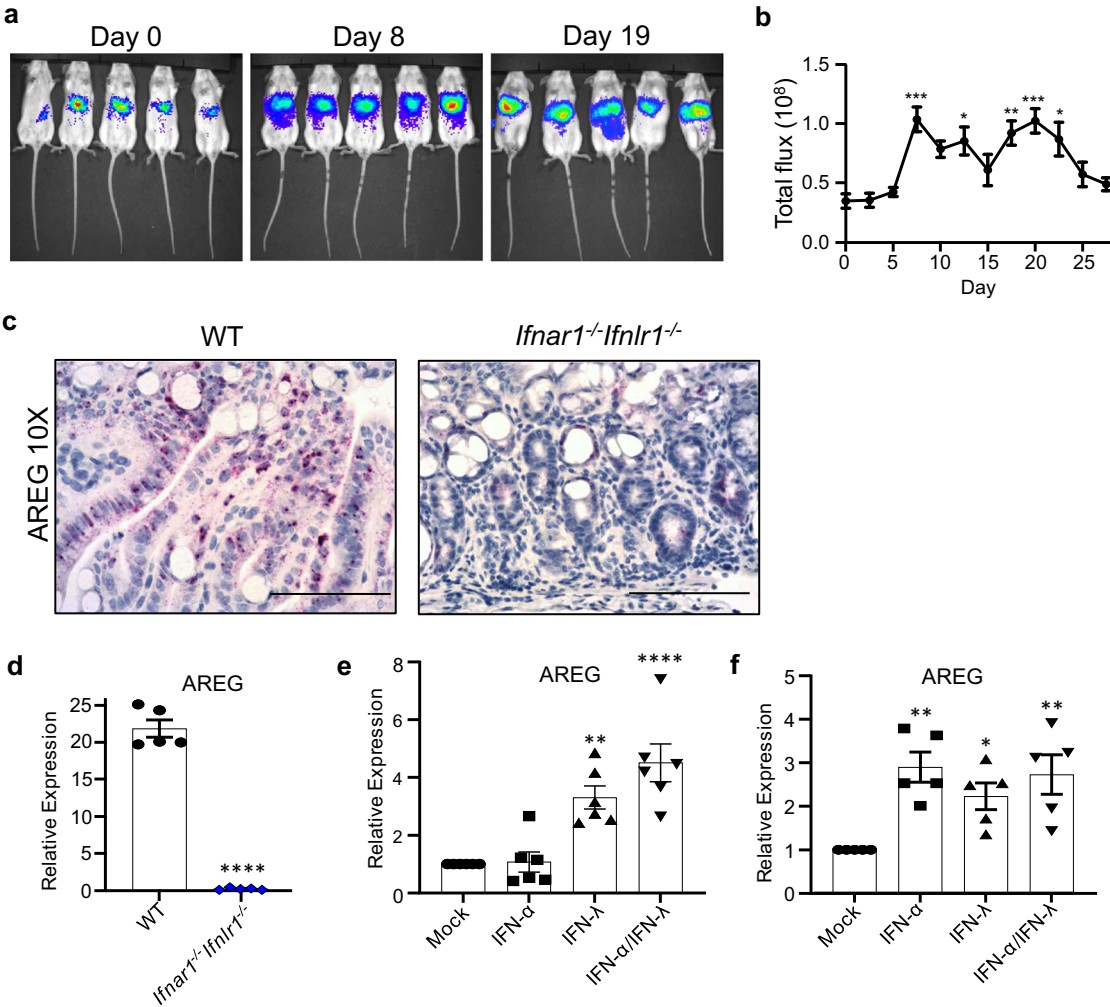

**Fig. 6 Ifnar1⁻/⁻Ifnlr1⁻/⁻ mice have diminished AREG expression following DSS treatment and IFNs regulate AREG expression. a, b** Mx2-Luciferase reporter mice (*n* = 5) were exposed to 1.5% DSS in drinking water for 7 days, followed by regular water. Mice were injected with luciferin and levels of IFN-controlled Mx2 promoter-driven luciferase expression were measured by IVIS on selected days. **c, d** 6–8-week-old WT and *Ifnar1⁻/⁻ Ifnlr1⁻/⁻* mice were administered 1.5% DSS in drinking water for 7 days, followed by regular water. Levels of colonic AREG transcripts were analyzed on day 11 by **c** in situ hybridization (*n* = 5) and **d** qPCR from total colon homogenates (*n* = 5). **e** Murine intestinal epithelial cells (mIECs) or **f** fluorescent-activated cell (FAC)-sorted bone marrow-derived CD45⁺ cells were treated with IFN for 30 min and upregulation of AREG transcripts were analyzed by qPCR (**e**, *n* = 6; **f**, *n* = 5). Scale bars are 100 μM. Data are representative of two (**a, b**) or three independent experiments (**c**) or pooled from two independent experiments (**d–f**). Symbols represent individual data points (**d–f**). Quantitative data were analyzed using **b, e, f** one-way ANOVA or **d** two-tailed unpaired *t*-test followed by Tukey's (**b**) or Bonferroni's multiple comparisons test (**e, f**) and represent mean values with SEM. **b** \**P* = 0.0231 (day 13), \**P* = 0.0168 (day 22), \*\**P* = 0.0055 (day 18), \*\*\**P* = 0.0004 (day 8), \*\*\**P* = 0.0005 (day 19); *P* values are for the indicated day compared with day 0. **d** \*\*\*\**P* ≤ 0.0001 and is for the indicated animal group compared with control group of WT mice. **e** \*\**P* = 0.0027, \*\*\*\**P* ≤ 0.0001. **f** \**P* = 0.0483 (IFN-λ), \*\**P* = 0.0023 (IFN-α), \*\**P* = 0.005 (IFN-α/IFN-λ). *P* values are for the indicated treatment compared with mock (**e, f**).

expression in mIECs by IFN-λ (Fig. 6e), demonstrate that IFN-λ can promote migration of epithelial cells through the engagement of EGFR signaling pathway.

To verify that impaired AREG expression contributes to the development of severe colonic ulceration in *Ifnar1⁻/⁻Ifnlr1⁻/⁻* mice following DSS-induced colitis, we conducted a rescue experiment and administered exogenous AREG to *Ifnar1⁻/⁻Ifnlr1⁻/⁻* mice on days 5, 7, 9, and 11 of the DSS treatment protocol. AREG treatment provided protection to 100% of *Ifnar1⁻/⁻Ifnlr1⁻/⁻* mice (Fig. 7a), and restored normal levels of colon epithelial cell proliferation as evaluated by Ki-67 IHC staining (Fig. 7b). These studies demonstrate that IFNs induced by tissue injury act to promote repair through the activation of EGFR signaling. Taken together, these data demonstrate that type I IFN signaling in the hematopoietic compartment and type III IFN signaling in the epithelial compartment supports mucosal healing via the upregulation of AREG expression. Loss of

both type I and type III IFN signaling results in AREG insufficiency following DSS treatment, impaired epithelial recovery, and the development of lethal disease. Supplying exogenous AREG can protect against these effects.

## Discussion

IFNs are multifunctional cytokines best known for their function in antiviral defense[26,54,55]. Although type I IFNs have long been considered to be the indispensable first-line mediators of the antiviral response, this view changed in 2003 when the IFN family was expanded by the discovery of type III IFNs or IFN-λs[12,56]. These newly identified cytokines, represented by four members in humans[12,57], and two members in mice[58], engage a unique IFN-λ-specific receptor complex to induce canonical type I IFN-specific signal transduction pathways independent of type I IFNs.

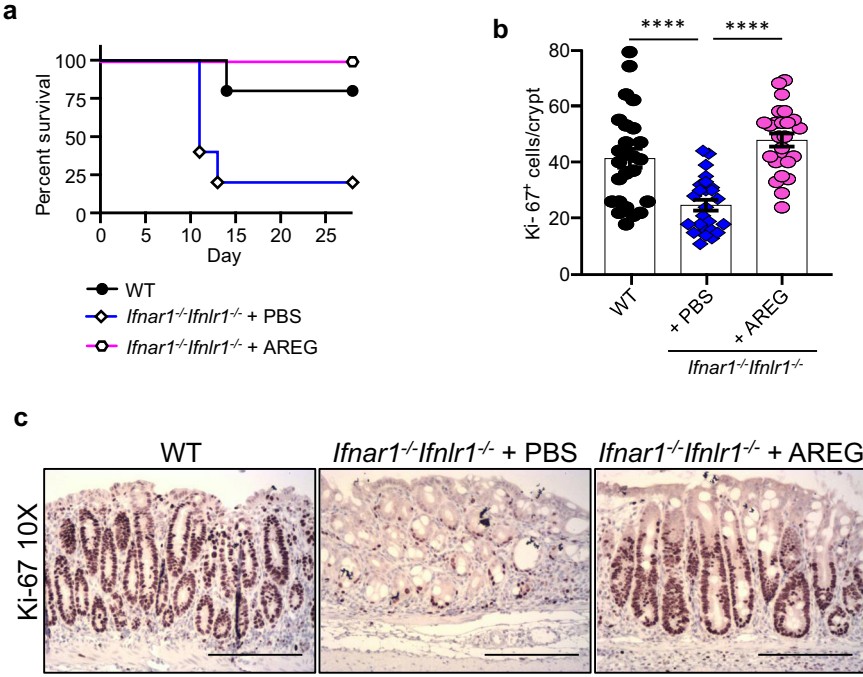

**Fig. 7 Administration of AREG rescues *Ifnar1*−/−*Ifnlr1*−/− mice from experimental DSS-induced colitis and restores colonic regeneration. a–c** 6–8-week-old WT and *Ifnar1*−/−*Ifnlr1*−/− were exposed to 1.5% DSS in drinking water for 7 days, followed by regular water. On days 5, 7, 9, and 11, *Ifnar1*−/−*Ifnlr1*−/− mice were administered AREG (5.0 µg) or PBS as a control by IP injection, and the animals were monitored daily for survival (**a**) (*n* = 5). **b**, **c** On day 11, mice were euthanized (*n* = 5) and proliferation of epithelial cells in the colon was evaluated by IHC staining for Ki-67. Symbols in **b** represent the number of Ki-67+ cells per crypt (five crypt/slide/animal; *n* = 5). **c** Representative images. Scale bars are 100 µM. Data are representative of two independent experiments. Quantitative data were analyzed using one-way ANOVA followed by Bonferroni's multiple comparisons test and represent mean values with SEM. ****$P \leq$ 0.0001.

Consequently, the biological activities of type I and type III IFNs are very similar, if not identical, in cells that respond to both types of IFN. All type I IFNs signal through the IFNAR1/IFNAR2 heterodimeric receptor, whereas signaling by type III IFNs is mediated through the IFNLR1/IL10R2 receptor complex. This engagement of distinct receptor complexes provides a molecular basis for the targeting of different cell subsets by type I and type III IFNs, depending on the pattern of IFNR expression. Moreover, overlapping, but nonidentical, production of type I and type III IFNs in response to the same stimuli[16,17,59,60] and recently reported differences in noncanonical IFN signaling pathways[52,59,61–64] are additional factors in their diverse biological functions in vivo.

Distinct contributions of type I and type III IFNs to the overall antiviral response is particularly well documented in the small intestine, where immune cells within the lamina propria respond only to type I IFNs, whereas epithelial cells in adult mice are exclusively sensitive to type III IFNs[15,17,18]. Of interest, epithelial cells of neonatal mice are responsive to both IFN types, indicating the existence of mechanisms regulating responsiveness of cells to IFNs[15]. Our current data (Fig. 5d) expand the observed compartmentalization of IFN responses to the colon, suggesting that the compartmentalized actions of type I and type III IFNs are likely to be preserved through the entire GI tract, at least in adult animals under homeostatic conditions. The exclusive ability of type III IFNs to act on IECs underlies their nonredundant protective functions against infections with GI viruses such as rotavirus[15,18], reovirus[17], and norovirus[65,66]. Moreover, the induction and action of type I and type III IFNs are not confined to virus infection, since bacterial as well as fungal infections also trigger IFN synthesis, and are in turn impacted by IFNs[16,60,67,68]. Based on these observations it is expected that IFNs would be involved in the regulation of GI homeostasis, particularly when mucosal barrier integrity is disrupted as occurs in IBD. Indeed, we observed activation of IFN signaling at day 8 of the DSS treatment protocol, which persisted for 2 weeks after DSS treatment was discontinued (Fig. 6a, b and Supplementary Fig. 10). These data are consistent with previous reports of IFN-λ production in the mouse model of DSS-induced acute colitis and in clinical samples from IBD patients[69].

The antiproliferative and proapoptotic properties of IFNs have been well characterized[70,71]. These activities are part of the antiviral defense mechanisms enabling infected cells to undergo apoptosis to prevent virus replication and halt virus spread. However, these IFN-mediated effects could also inhibit maintenance of the epithelial barrier, particularly in the intestinal tract with its high rate of epithelial cell turnover. Typically, IFNs alone do not initiate apoptosis, but act to sensitize cells to other triggers of apoptosis such as infection. IFN-enhanced epithelial apoptosis may be beneficial for controlling pathogen spread, but could also antagonize the rapid restoration of barrier function required to maintain homeostasis. Actions of IFNs in this respect may be reminiscent of the dual function of inflammation in tissue homeostasis[72]. On the one hand, inflammation is a cause of tissue injury, but at the same time inflammatory cells and cytokines contribute to infection control, clearance of cell debris and initiation of tissue regeneration and repair. Several proinflammatory cytokines, including IL-6 and TNF-α, have been shown to participate in reparative processes[72–77]. These cytokines promote epithelial regeneration by acting directly on IECs and by upregulating expression of EGFR ligands, including AREG[77–80]. Our current study of DSS-induced acute colitis revealed that, similar to the dual function of these proinflammatory cytokines, type I and type III IFNs appear to act in a balanced and

compartmentalized fashion to control infection while mitigating tissue damage by the simultaneous promotion of tissue repair.

Although it is now well-appreciated that tonic activation of innate receptors by microbial byproducts is important for GI homeostasis[81,82], the downstream mediators are not well characterized. TLR-mediated signaling pathways regulate expression of numerous cytokines, including type I and type III IFNs. IL-22, IL-33, and thymic stromal lymphopoetin have all been shown to promote recovery from DSS-induced colitis[9–11,83,84] as have TLR3 and TLR9 agonists[85,86]. Conversely, deficiencies in MYD88, TLR2, TLR4, and IRF3 render mice more susceptible to DSS-induced injury[84,87–89]. Taken together, these data suggest that IFNs may be involved in the regulation of GI homeostasis downstream of innate receptors, but published studies with mouse models of DSS-induced colitis have not yet generated consistent results. On the one hand, $Ifnar1^{-/-}$ mice are more sensitive to DSS-induced colitis, and administration of type I IFNs is protective[90,91], but recovery from DSS-induced injury was accelerated in IFNAR-deficient mice[21], and intestinal delivery of IFN-β by $Lactobacillis$ exacerbated the DSS-induced symptoms[92]. Clinical studies have also been inconclusive, reporting beneficial or detrimental outcomes, or no effect of type I IFNs, in IBD patients[93,94]. Protective activities of type III IFNs in DSS-induced colitis, suppressing intestinal inflammation and promoting mucosal healing, were recently reported[20,63,69], but the use of different DSS treatment protocols by laboratories makes comparison between the studies challenging.

The acute inflammatory response has been the focus of much UC research, and the significance of IFN signaling during this phase has been demonstrated[20,21,63,69]. Exposure of mice to high concentrations of DSS, between 2 and 3%, resulted in a robust inflammatory response and severe damage to the colonic epithelium. The use of a lower concentration of DSS (1.5%) in our studies allowed us to evaluate deficiencies in mucosal repair in IFNR-deficient mice. Under these conditions, we observed a marked susceptibility of $Ifnar1^{-/-}Ifnlr1^{-/-}$ mice to DSS treatment, relative to single IFNR-deficient or WT mice (Fig. 1a–e). However, increased colon infiltration of neutrophils and other immune cells was detected in all IFN-deficient mice (Fig. 3a) consistent with previous reports that type I and type III IFNs suppress neutrophil recruitment[16,95,96]. Type III IFNs act directly on neutrophils, but the response of this cell type to IFNs varies depending on the disease studied[16,63,96–98]. In our model we observed that, despite the enhanced neutrophil recruitment (Fig. 3a), IFNR-deficient mice had dampened neutrophil function as measured by the reduction of MPO activity (Fig. 3b). This impaired neutrophil activation observed in $Ifnar1^{-/-}$, $Ifnlr1^{-/-}$, or $Ifnar1^{-/-}Ifnllr1^{-/-}$ mice is consistent with our previous observation that responsiveness of neutrophils to both type I and type III IFNs is required for efficient antifungal killing and ROS production[16]. We also observed that levels of LCN-2 in feces of WT and IFNR-deficient mice, a measure of intestinal inflammation[36], changed with similar kinetics (Fig. 3c). Overall, although immune cell recruitment to the injured colon was augmented in IFNR-deficient mice, histology, MPO activity, and LCN-2 secretion did not suggest that increased inflammation was responsible for the enhanced sensitivity of $Ifnar1^{-/-}Ifnlr1^{-/-}$ mice to DSS treatment.

We hypothesized that impaired regeneration of the colonic epithelium following DSS treatment could explain the enhanced susceptibility of $Ifnar1^{-/-}Ifnlr1^{-/-}$ mice. In support of this hypothesis, diminished colonic epithelial proliferation following injury was observed in $Ifnar1^{-/-}Ifnlr1^{-/-}$ and $Ifnar1^{-/-}$ mice, with $Ifnar1^{-/-}Ifnlr1^{-/-}$ being most dramatic (Fig. 4). The EGFR signaling pathway has been identified as a critical mediator of epithelial regeneration[77–80,99], and there was a significant

decrease in the expression levels of the EGFR ligand AREG in $Ifnar1^{-/-}Ifnlr1^{-/-}$ mice following DSS exposure (Fig. 6c, d). Importantly, administration of exogenous AREG to $Ifnar1^{-/-}Ifnlr1^{-/-}$ mice abrogated the lethal effects of DSS administration and restored the proliferative capacity of the colonic epithelium to WT levels (Fig. 7a–c). DSS treatment of bone marrow chimeras revealed that either type I IFN signaling in the hematopoietic compartment or type III IFN signaling in the epithelial compartment was sufficient to support epithelial regeneration in the large intestine (Fig. 5a, c). This contribution by either compartment was mediated by upregulation of AREG expression in CD45$^+$ cells by type I IFNs or in epithelial cells by type III IFNs (Fig. 6e). While the absence of type I and type III IFN signaling is tolerated at steady state (Supplementary Fig. 8), the proliferative burst required for re-epithelialization is impaired in $Ifnar1^{-/-}Ifnlr1^{-/-}$ mice. It is of interest that $Ifnar1^{-/-}Ifnlr1^{-/-}$ mice treated with AOM/DSS to drive inflammation-induced CRC had an elevated tumor burden (Supplementary Fig. 2b), an apparent discrepancy given the decreased proliferative capacity of the colonic epithelium of these animals after DSS treatment (Fig. 4). However, this apparent contradiction may be explained by the increased tumor burden in mice deficient in EGFR signaling[100,101]. In the AOM/DSS model, EGFR signaling acts as a tumor suppressor by stimulating mucosal regeneration, thereby limiting the duration of the inflammatory response[100,101]. In that setting, delayed repair and chronic inflammation was associated with increased tumorigenesis.

Based on our data, several mechanisms can be proposed to explain how AREG promotes re-epithelialization and regeneration of intestinal epithelium. AREG is initially produced as a transmembrane protein that can activate EGFR signaling in a juxtacrine fashion in adjacent cells[45]. AREG can also be proteolytically cleaved from the membrane and in its shed form act as a paracrine signal[45]. Our data demonstrate that IFN-λ stimulates migration of mIECs in an EGFR-dependent manner (Supplementary Fig. 14), an activity that promotes re-epithelialization. In situ hybridization shows the presence of AREG transcripts in cells surrounding the sites of DSS damage (Fig. 6c), where AREG would act locally to promote epithelial migration to preserve barrier integrity. EGFR is highly expressed by mature terminally differentiated enterocytes[102] that are constantly shed and replenished by the process of IEC turnover. Therefore, IFN-λ-driven upregulation of the membrane-bound form of AREG by cells at the tips of villi and at the luminal surface would accelerate re-epithelialization of the damaged areas. EGFR signaling also regulates proliferation and differentiation of stem cells in the crypts[103,104]. "Reserve" or "quiescent" intestinal stem cells (ISCs) give rise to LGR5+ crypt base columnar stem cells (CBCs). Both ISCs and CBCs are located at the bottom of the crypts. CBCs then differentiate into "transient-amplifying" enterocyte progenitor cells and secretory progenitor cells, which are positioned at the crypt sides[103]. Of note, in $Ifnar1^{-/-}Ifnlr1^{-/-}$ mice following DSS exposure, Ki-67$^+$ cells are still present at the base of the crypts but staining for Ki-67, a marker for cell proliferation, is lost in the progenitor cell compartment and above, as it is observed in WT or single IFNR-deficient mice (Fig. 4b). Therefore, in the absence of IFN signaling, generation and differentiation of CBCs appear to be impaired. Of interest, "reserve" ISCs express leucine-rich repeats and immunoglobulin-like domains 1 (Lrig1) protein, a negative regulator of EGFR signaling[105–107]. Lrig1 expression is suppressed by IFN signaling (Supplementary Data 1). Thus, IFN signaling not only stimulates AREG upregulation but also downregulates Lrig1 expression in "reserve" ISCs, events which should stimulate cycling of ISCs, and accelerate replenishment of the CBC compartment and subsequently of the entire epithelium. Noticeably, AREG treatment of $Ifnar1^{-/-}Ifnlr1^{-/-}$ mice restores Ki-67

positivity above the bottom of the crypts (Fig. 7c), suggesting that IFN-dependent upregulation of AREG stimulates proliferation of progenitors and migration of IECs along the crypt. A recent publication suggests an additional repair mechanism in which AREG-triggered signaling induces local release of bioactive TGF-β from mesenchymal stromal cells that in turn stimulates stem cell differentiation[108]. In addition to stromal cells, EGFR signaling in myeloid cells has also been shown to support proliferation of IECs through the upregulation of IL-6[42]. Therefore, during repair process in the absence of IFN-induced AREG upregulation, repopulation of the damaged epithelium is compromised by the lack of sufficient stimuli for differentiation, proliferation, or migration of IECs.

Overall, our study revealed an unexpected role for type I and type III IFNs in promoting epithelial regeneration following intestinal injury. This novel function of IFNs at the intestinal mucosal surface is driven by their ability to upregulate AREG, thus promoting restoration of the epithelial barrier. By enhancing repair at the mucosal surface, type I and type III IFNs attenuate DSS-mediated colitis, and may therefore represent a new treatment option for inflammatory conditions such as IBD.

## Methods

**Mice.** Conventional specific pathogen-free (SPF) WT and congenic CD45.1 WT mice were purchased from Jackson Laboratory (stock nos 000664 and 002014, respectively). $Ifnar1^{-/-}$, $Ifnlr1^{-/-}$, and $Ifnar1^{-/-}Ifnlr1^{-/-}$ mice were generated in the laboratory as previously described[15]. Transgenic Mx2-Luc reporter mice with intact type I and type III IFN receptors were obtained from Hansjörg Hauser and Mario Köster[43]. All mouse strains used in these studies were on C57BL/6 background and housed in individually ventilated M.I.C.E. cages (Animal Care Systems Inc.) with Lab Grade Aspen Shavings (NEPCO) at the same location at the SPF barrier facility at NJMS, Rutgers University, and all protocols were approved and permission to perform animal experiments was granted by Rutgers University (Newark) Institutional Animal Care and Use Committee. Mice were kept on a standard 12 h light-dark cycle at 20–22 °C and 30–70% humidity with access to food (Laboratory Autoclavable Rodent Diet 5010, LabDiet) and autoclaved reverse osmosis water ad libitum. All mouse strains were housed separately (except for cohousing experiments described below) and bred as homozygotes, and mice for the experiments were weaned at day 21 postpartum, separated based on gender, and kept in individual cages (≤5 mice per cage).

**AOM/DSS administration.** For all experiments, 6–8-week-old mice of various strains were age and gender-matched and given a single IP injection of AOM 12.5 mg/kg body weight. Animals were subsequently exposed to three cycles of 1.5% DSS (weight/volume) in drinking water for 7 days, with 14 days of sterile water between each cycle for recovery. On day 90, animals were euthanized and gross tumor burden was evaluated. Mice of each strain and gender were kept in separate cages. Animals who lost 25% of their body weight were euthanized by $CO_2$ inhalation according to protocol.

**DSS administration and disease score.** For all experiments, mice of various strains were age and gender-matched and administered 1.5% DSS (weight/volume) in drinking water for 7 days and subsequently placed on regular drinking water. Mice were 6–8 weeks old at the beginning of DSS treatment, except experiments with bone marrow chimeras, where mice were 6–8 weeks old at the time of bone marrow transplantation. The recipient mice underwent recovery for 8 weeks prior to DSS treatment. Therefore, control mice for experiments with bone marrow chimeras were also aged to 14–16 weeks to match the age of the chimeras. Mice of each strain and gender were kept in separate cages, except for cohousing experiments. For cohousing experiments, age-matched WT and $Ifnar1^{-/-}Ifnlr1^{-/-}$ female mice were weaned at 21 days and cohoused right after in sterile cages at a ratio of 1:1 (WT:$Ifnar1^{-/-}Ifnlr1^{-/-}$; four mice per cage) or 2:3 (WT:$Ifnar1^{-/-}Ifnlr1^{-/-}$ or $Ifnar1^{-/-}Ifnlr1^{-/-}$:WT; five mice per cage). Male mice were weaned at 21 days and bedding from either WT or $Ifnar1^{-/-}Ifnlr1^{-/-}$ cages was added to their counterpart's cages. Cohousing or cross-bedding was conducted till the start of DSS treatment when mice were 6–8 weeks old and continued till the end of the experimental procedures. Mice were monitored daily for weight loss, morbidity, and clinical disease (0—normal, 1—soft feces, 2—diarrhea, 3—diarrhea, visible bleeding).

**Tissue harvesting, fixation, and immunohistochemistry.** Large intestine was removed from cecum to anus, opened longitudinally, and washed three times with ice-cold 1X PBS. H&E and PAS/Alcian Bblue staining was performed by the Histology Core Facility on formalin-fixed, paraffin-embedded (FFPE) large

intestine tissue slides (Rutgers NJMS). Staining for Ki-67 and pSTAT1 was performed on 5-μm sections of FFPE large intestine tissue. Tissue was rehydrated and treated for antigen unmasking. Sections were incubated for 10 min in hydrogen peroxide, washed with PBS, and blocked for 5 min with Super Block (ScyTek). Sections were washed in 0.05% Tween-20 in PBS (PBST (0.05%)), and incubated overnight with rabbit monoclonal Ki-67 antibody (clone SP6, Fisher Scientific, ready to use) or rabbit monoclonal phosphoTyr701-STAT1 (clone 58D6, Cell Signaling, 1:500 dilution). Slides were washed in PBST (0.05%), and incubated for 30 min with UltraTek anti-rabbit biotinylated antibody (ABK, ScyTek; ready to use) at room temperature. Sections were washed in PBST (0.05%) and incubated with UltraTek Streptavidin/HRP (ScyTek) for 20 min at room temperature. Sections were washed and NovaRED substrate solution (Vector) was used as a substrate. Slides were subsequently washed and counterstained with Mayer's haematoxylin and Scott's bluing buffer.

**Analysis of inner mucus layer.** Six-to-eight-week-old naive DSS-untreated mice were euthanized. Colon was removed, opened longitudinally, and gently washed once with ice-cold 1X PBS to remove feces. Tissue was fixed in Carnoy's solution for 4 h at room temperature and subsequently moved to ethanol. PAS/Alcian blue staining was then performed on sections of Carnoy's solution-fixed, paraffin-embedded tissue slides and inner mucus layer thickness was analyzed using ImageJ 1.x (NIH).

**Protein quantification.** Fecal pellets were collected on select days in PBST (0.1%) and stored until assay. Samples were diluted 1:100 and ELISA was conducted according to the manufacturer's recommendation (R&D Systems). Optical density was measured using a plate reader, and sample concentrations were calculated by regression analysis.

Total colon homogenates were collected on select days and samples were assayed using V-PLEX Plus Proinflammatory Panel 1 Mouse Kit and V-PLEX Mouse IL-22 Kit (Meso Scale Diagnostics, MSD) according to the manufacturer's instructions. Briefly, calibrator dilutions, controls, buffers, and plates were prepared according to instructions. Diluted samples with equal protein concentrations were added to each well and left at room temperature with shaking for 2 h. The plates were washed three times and detection antibody solution was added and incubated at room temperature with shaking for 2 h. Read buffer was added to the plates and analytes of interest were measured using MS2400 imager (MSD), and concentrations were extrapolated based on the standard curves.

**Luciferase detection in Mx2-Luc reporter mice.** Transgenic Mx2-Luc reporter mice, which are WT for the type I and type III IFN systems[43], were exposed to 1.5% DSS in drinking water for 7 days, followed by regular water, and the levels of Mx2 promoter-driven luciferase expression were assessed by measuring luciferase activity (bioluminescence emission) by whole body imaging of live animals administered with luciferin as described[43]. For in vivo imaging, mice were anesthetized on select days of the DSS treatment protocol by isoflurane inhalation and IP-injected with 15 μg/μl of luciferin (Promega) in 100 μl of 1X PBS. Mice were placed in the imaging chamber of a Xenogen IVIS (IVIS 200, Xenogen) for 20 min with luminescent images being acquired every minute. Signal emission from the entire luminescent area of each mouse was measured and expressed as photon flux (photons count per second; p/s), and the photon fluxes at the peak plateau were recorded.

**MPO in vivo imaging.** On day 8 from the start of DSS treatment, neutrophil infiltration and function was assessed as described[109]. Briefly, the MPO tracer L-012 (Wako Chemical) was dissolved to a concentration of 20 mM in sterile water. One hundred microliters of the L-012 solution was administered via IP injection. Animals were anesthetized and luminescence was quantitated using IVIS 200 as described above for experiments with Mx2-Luc mice. Animals were subsequently euthanized and neutrophil infiltration was quantitated by flow cytometry. MPO activity was reported relative to number of colon-infiltrating neutrophils (photon flux (p/s) from the entire luminescent area of each mouse divided by number of colon-infiltrating neutrophils).

**Enumeration of innate immune cell infiltrates by flow cytometry.** Large intestine samples were removed, open longitudinally, and gently washed with ice-cold PBS. Tissue was cut into small pieces and washed twice in predigestion solution (1X HBSS, 5 mM EDTA, 1 mM DTT) for 20 min at 37 °C with gentle shaking. Tissue was then enzymatically digested in collagenase D (0.05 g/100 ml PBS). Collected fractions were pooled within sample and lysed to remove red blood cells. Staining of single-cell suspensions included the following antibodies from BD: Ly6C (FITC; clone AL-21), CD11b (PerCP-Cy5.5; clone M1/70), Siglec F (BV421; clone E50-2440), CD11c (Alexa Fluor 700; clone HL3), CD45.2 (BUV395; clone 30-F11), EpCAM (PE; clone G8.8), Ly6G (PE-Cy7; clone 1A8), and MHCII (BV711; clone M5/11.415.2). CD169 (eFluor 660; clone SER-4) was purchased from eBioscience, and F4/80 (APC/Cy7; clone BM8) was purchased from BioLegend. DAPI (Life Technologies) was used as a viability control. Samples were collected on BD LSRFortessa X-20 with the use of BD FACSDiva v.8.0 software for

data acquisition, analyzed using FlowJo v.10.2 software, and enumerated by back calculating based off of percentages relative to total cell counts.

**RNA in situ hybridization**. RNA in situ hybridization for AREG was performed according to the manufacturer's instructions (ACD). Briefly, FFPE slides were baked for 1 h, followed by deparaffinization. Slides were treated with pretreat 1, washed, and subsequently treated with pretreat 2. Slides were then washed and treated with pretreat 3. AREG probe (430501) was added to slides for 2 h, followed by signal amplification and detection. Probed slides were subsequently counterstained and mounted.

**Generation bone marrow chimeras and host engraftment**. Bones from the hindquarters of 6–8-week-old mice were flushed with PBS to remove cells, and cell suspensions underwent RBC lysis. Donor-derived lymphocyte-depleted bone marrow cells were adoptively transferred into lethally irradiated 6–8-week-old recipient mice via IV injection. Recipient mice were placed on augmentin-supplemented water for 4 weeks, followed by regular water for 4 weeks. Prior to use in experiments, bone marrow chimera mice donor engraftment of the hematopoietic compartment was analyzed by flow cytometry with CD45.1/CD45.2 antibodies performed on peripheral blood cells CD45.1 (FITC, clone A20, BD), CD45.2 (BUV395, clone 30-F11, BD), and DAPI.

**Treatment of cells with IFN, RNA isolation, and qPCR**. A clonal population (clone G7) of immortalized mIECs (mIEC-1) obtained from Tobias May[46] was selected based on its high transepithelial electrical resistance > 3000 $\Omega/cm^2$ when grown as polarized culture in transwells. For qPCR analysis, mIECs (mIEC-1 clone G7, mIEC-G7), murine lung epithelial cells (MLE-15, ATCC), and FAC-sorted CD45$^+$ cells were treated for 3 h with 100 ng/ml of respective IFN (recombinant mouse IFN-α2, Novoprotein; recombinant mouse IFN-λ2, either from Peprotech or Bristol-Myers Squibb (pegylated or nonpegylated forms)). RNA was extracted using QIAshredder according to the manufacturer's instructions. For AREG expression analysis from total colon tissue of DSS-treated mice, animals were euthanized on day 11 from the start of DSS treatment and the large intestine was homogenized in 1.0 ml of Trizol reagent; RNA was extracted according to the manufacturer's instructions. One microgram of total RNA was reverse-transcribed to cDNA using high capacity cDNA reverse transcription kit (Applied Biosystems). TaqMan (Applied Biosystems) probes and associated master mix were used for qPCR. Each gene was normalized to GAPDH and gene expression was calculated using the ΔΔCT method relative to naive samples. For mock sample comparisons, the ΔCT between GAPDH and AREG was graphed.

**IECs isolation and RNA sequencing**. Six-to-eight-day-old WT mice were treated daily for 3 days with mIFN-α or mIFN-λ (500 ng per intramuscular injection) and live EpCAM$^+$ (CD45$^-$CD326$^+$) IECs were FAC-sorted into separate samples from 3 mice per treatment group (three independent biological replicates). Total RNA was isolated from the cell samples with the use of Qiagen RNeasy Mini Kit. The quality of the RNA was determined with Bioanalyzer 2100 (Agilent). Samples with RNA integrity number > 7.0 were used for subsequent processing. Total RNA was subjected to two rounds of poly(A) selection using oligo-d(T)25 magnetic beads (New England Biolabs). Illumina compatible RNA-seq library was prepared using NEB next ultra RNA-seq library preparation kit. The cDNA libraries were purified using Ampure XP beads and quantified on an Agilent Bioanalyzer and qubit analysis. Equimolar amounts of barcoded libraries were pooled and sequenced on Illumina NextSeq 500 platform (Illumina, San Diego, CA) with 1 × 75 configuration. Raw reads were quality trimmed using Trimmomatic-0.33 with leading and trailing Q score 25, minimum length 25 bp, and adapters were further removed using FASTX clipper (FASTX-Toolkit/0.0.14). The cleaned reads were mapped to *Mus musculus* genome GRCm38 using Tophat v.2.0.13. The reference genome sequence and annotation files were downloaded from ENSEMBLE, release.83 (Mus_musculus.GRCm38.83.fa and Mus_musculus.GRCm38.83.gtf). The aligned read counts were obtained using htseq-count as part of the package HTSeq-0.6.1. The Bioconductor package edgeR (version 3.8.6 with limma 3.22.7) was used to perform the differential gene expression analysis, under R environment, R version 3.1.2. Raw FASTQ files for the RNA-seq libraries have been deposited in the NCBI Sequence Read Archive (SRA) and assigned BioProject accession number PRJNA579563 with Biosample accession numbers SRR10968324, SRR10968325, SRR10968326, SRR10968327, SRR10968328, SRR10968329, SRR10968330, SRR10968336, and SRR10968337 with the link in the NCBI database (https://www.ncbi.nlm.nih.gov/bioproject/PRJNA579563).

**AREG administration**. Six-to-eight-week-old mice were treated with 1.5% DSS as described above. Mice were treated with 5.0 μg of AREG (Biolegend) or PBS on days 5, 7, 9, and 11 via IP administration in 300 μl. Mice were monitored daily for survival and morbidity.

**Cell proliferation and migration assays**. For cell proliferation assay, immortalized mIECs (mIEC-G7) were plated on six-well tissue culture plates at 40% confluency (4 × 10$^5$ cells per well) and cultured in RPMI media supplemented with

10% FBS without any treatment or in the presence of recombinant mouse IFN-λ2 (100 ng/ml) or AREG (100 ng/ml). Cells were collected at 24 and 48 h and numbers of live cells were counted by Vi-Cell cell analyzer (Beckman Coulter).

For scratch migration assay, the mIEC-G7 cells were grown to confluency in 12-well plates and a single scratch per well was made with a 1 ml sterile tip. Cells were subsequently cultured in RPMI media supplemented with 10% FBS without any treatment or in the presence of recombinant mouse IFN-λ2 (100 ng/ml) or AREG (100 ng/ml) with or without EGFR tyrosine kinase inhibitor Gefitinib (500 nM; Selleckchem). Images were taken immediately after the scratch was made and at 24 h, and width of each scratch was measured in pixels at four randomly positioned points. The distance migrated by the cells within 24 h was calculated as half of the difference between gap width at 0 and 24 h.

Real-time cell migration assay was performed as described[110] using the XCELLigence RTCA DP instrument (Agilent). Briefly, the mIEC-G7 cells were serum starved for 8 h in RPMI medium supplemented with 0.5% FBS and then seeded in the same media (4 × 10$^5$ cells in 100 μl per well) in the upper chamber of XCELLigence RTCA DP CIM-16 plate. One hundred and eighty microliters of RPMI media supplemented with 10% FBS was added in the lower chamber of the wells, and the cells were left untreated or treated with pegylated murine IFN-λ2 (100 ng/ml) or murine AREG (16.67 ng/ml) added in the upper chamber. Changes in cell index depicting cellular migration to the bottom chamber were assessed every 10 min for 28 h.

**Statistical analyses**. Most data, unless otherwise stated in the figure legends, present results of a single independent experiment demonstrating mean values with SEM. All experiments were independently repeated at least twice. Data comparing two groups were analyzed by unpaired two-tailed *t*-test followed by Bonferroni or Tukey's post hoc test. Data analyzing more than two groups were analyzed by one-way or two-way analysis of variance (ANOVA) followed by Bonferroni or Tukey's post hoc test or by nonparametric Kruskal–Wallis test followed by Dunn's post hoc test as specified in figure legends. The $P$ value < 0.05 indicated statistically significant differences. Specific $P$ values are presented for each experiment in figure legends. GraphPad Prism 9.0.0 was used for data analysis.

**Reporting summary**. Further information on research design is available in the Nature Research Reporting Summary linked to this article.

## Data availability
The data that support the findings of this study are available from the corresponding author upon request. RNA-seq data have been deposited in the NCBI SRA with the accession number PRJNA57956. All other data are included in the Supplementary Information or available from the authors upon reasonable requests, as are unique reagents used in this article. Source Data are provided with this paper.

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

## Acknowledgements

The authors are thankful to Hansjörg Hauser and Mario Köster for Mx2-Luc reporter mice, Tobias May for immortalized mIEC (mIEC-1) line, Sean Doyle and Bristol-Myers Squibb for providing pegylated or nonpegylated forms of recombinant mouse IFN-λ2, Luke Fritzky and Histology Core Facility for the help with tissue slides and imaging, and Jerome Langer, Nan Gao, and Karen Edelblum for critical reading and suggestions. This work was supported in part by National Institutes of Health Grants RO1 AI104669 (to S. V.K. and J.E.D.) and R01AI114647 and a Burroughs Wellcome Investigators in the Pathogenesis of Infectious Disease award (to A.R.), New Jersey Commission on Cancer Research fellowship (to C.M.), and New Jersey Health Foundation (to S.V.K.).

## Author contributions

Conceived and designed the experiments: C.M., J.E.D., and S.V.K. Performed the experiments: C.M., V.E., J.-D.L., J.P., R.S., O.D., H.-C.T., S.V.S., H.R., M.J.S., and V.D. Analyzed and interpreted the data: C.M., Y.-J.C., B.P.P., R.B.B., M.G., A.R., J.E.D., and S. V.K. Wrote the paper: C.M., J.E.D., and S.V.K.

## Competing interests

S.V.K. is an inventor on patents and patent applications related to IFN-λs, which have been licensed for commercial development. Other authors declare no competing interests.
