## [Peer Review File · Nature Communications]

Reviewers' comments:

Reviewer #1 (Gut immunity, mucosal barrier)(Remarks to the Author):

McElrath et al have induced colitis by DSS in IFN receptor *Ifnar1* and *Ifnlr1* knock out animals. These double knock-out animals get more severe colitis, whereas the single ones are close to WT. Most importantly, transfer of WT cells to the double knock-out limits the disease. Furthermore, amphiregulin IP injections four times, something that rescued the animals.

This manuscript contains potentially interesting results, especially the beneficial effect of amphiregulin. However, the results and conclusions does not make it up to a consistent understanding and explanation for the observations. The authors introduce and discuss the role of the epithelium, but that bone marrow transplant from WT to the double knock-out reverts the colitis phenotype elegantly rule out any role of the epithelium. The results of Fig. 5A suggests that it is the *Ifnar1* cells that are important for the hematopoietic cells to decrease colitis, but instead of following this up, the author go to the epithelium (Fig. 5D). A lot of information is lacking. How are the IFN responding cells affecting the epithelium? How is the interesting amphiregulin effects mediated and linked to cells responding to IFN?

There are number of questions (and ways of presenting the results) that need to be addressed to make this understandable. The conclusions in the last sentences (line 458) that IFN promote epithelial generation and that this is mediated by amphiregulin require further consideration and experimentation.

Additional MAJOR

1. The authors claim that the double KO show goblet cell depletion. In Fig. 2B mucus plugs can be observed in two crypts, suggesting what is the typical situation and misunderstanding in colitis, a faster goblet cell emptying. This has to be addressed by staining for precursor forms of typical goblet cell proteins like *Muc2* and *Clca1*.
2. The claimed data showing mucus staining is not included although mentioned in text and figure legend S3.
3. Figures lack magnification bars and seems to be shown with different magnifications.
4. The staining for Ki-67 cells seems to suggest fewer such cells in the double KO, but the figure is of lower magnification. Even if this is the case, this does not show decreased proliferation and that this is 'responsible for the near total mortality' (line 210).
5. The RNA Seq data claimed (line 290) is not included. Only a Fig. S10. The raw data must be presented or at least the parts that related to AREG. Any difference of the two tested IFNs. Proof that the analyzed cells were the epithelial cells? PCR of total homogenate is not a good control for expression in epithelial cells.
6. Line 564. There is no information what type of AREG and supplier was used. This is a transmembrane protein, did they use the extracellular part.

Reviewer #2 (Colitis, gut immune responses)(Remarks to the Author):

General comments

McElrath et al. investigated whether the combined loss of type I and type III IFN signaling might enhance the susceptibility to DSS-induced colitis. For this purpose, the authors assessed mice lacking both types of IFN receptors and reported a marked loss of goblet cells and a diminished proliferation of epithelial cells in the colon. Furthermore, McElrath et al. reported that impaired mucosal healing in

double-knockout mice is driven by decreased AREG expression, which can be up-regulated via type I and type III IFN signaling. By using bone marrow chimera experiments, they also demonstrated that IFN signaling, in either the epithelial or hematopoietic compartments, is sufficient to provide protection against DSS-induced injury and to support epithelial cell proliferation in the intestinal mucosa. Collectively, data suggests a pleiotropic function for IFN signaling in colonic epithelial regeneration following DSS-induced damage. This manuscript is well written and contains novel and also potentially clinically relevant data. Experiments are generally well-planned and with some few exceptions, they seem to be well conducted. However, this referee found some critical points in this manuscript that need an extensive and careful revision by the authors, including the need to perform new experiments, before its acceptance for publication.

Major comments

One of my major concerns about the manuscript is the very condensed method's section. More recently, there is a growing number of publications, notably in high impact journals, about the lack of reproducibility of pre-clinical studies. This has led a substantial change in the guidelines for manuscript submission in most relevant international journals, including those of the Nature group. This manuscript is not in compliance with such recommendations. For instance, the authors must provide details on the animals used: species, strain, gender, age, and body weight. Please, also provide general information about the KO mice, especially regarding the genetic background, housing and husbandry conditions (the type of cage, bedding material, breeding programming, type of food, access to food and water, environmental enrichment, etc). Another very relevant point regards the sample calculation. The authors reported different numbers of animals per group. This great variation in the number of animals per group might cause serious problems and eventually bias in the statistical analysis of the data. Are the authors convinced that the small number of animals used in some experiments is sufficient to confirm that the observed statistical differences have a biological relevance? Furthermore, to avoid any bias of data interpretation, it is important to inform whether the in vivo experiments were conducted blindly (if the experimenter conducted the experiments without prior knowledge of treatments and mouse groups). Also important, were the animals assigned randomly to the various experimental groups? There are many literature examples showing that experiments carried out without randomization of animals in the different experimental groups produce serious bias in the analysis, consequently affecting data interpretation and the main conclusions of the study. If the animals had been previously randomized before experiments, please provide the method of randomization used. These above-raised points are essential to allow the reproducibility and the robustness of preclinical scientific papers. Indeed, many scientific journals have pointed out their critical importance and have made them mandatory for publications containing in vivo experiments. For more details, please, see the following references: Landis et al ; Nature 490, 187, 2012; Glenn and Ellis, Nature, 483, 531, 2012; Peers et al., Nature Review Drug Discovery, 11, 273,2012; Collins et al., Nature, 505, 612, 2014; Allison et al., Nature, 530, 27, 2016; Nature, 542, 409, 2017; Science 355, 234, 2017.

Other comments

1. Please, provide which was the initial experimental number of animals in each group. This was not described in the methods section.
2. What means the untreated DSS group? Please, provide data of animals that did not receive the DSS solution (naïve mice). This is important for purposes of comparison with the DSS-treated animals, and also to confirm the intestinal mucosa integrity and health of naïve animals.
3. The histological score is a non-parametric data; thus, the analysis of variance cannot be applied. Please, use a non-parametric test to analyze this data (Fig. 1E and 2 D).
4. I suggest to remove the figures 1 A and 1 B from the body of the manuscript and to transfer both figures to supplementary data, as they show that association of DSS and the carcinogen azoxymethane resulted in the death of the majority of animals.

5. Figure 2D is quite confused. I suggest to remove it and to describe the corresponding data in the results section.
6. In figure 4A, it appears that animals did not receive a DSS solution and in figure 4B, animals were treated with DSS. Please clarify these points in the figure legend.
7. Authors, please, check in the figures 1A and 1C if the ANOVA test was really applied since survival curves are depicted.
8. The authors represent the figures with Mean \pm Standard Error of the Mean (SEM), but the ARRIVE (Animal Research: Reporting of In vivo Experiments) guideline recommends the use of Standard Deviation (SD).
9. In figure 4B, the representative panel for *Ifnar1* knockout mice seems to present more intense labeling than that observed for WT or *Ifnlr1* mice. However, the quantification of the data shows different results. Please check this point.
10. Please, explain why the animals of the *Ifnar1* group show almost an equal proliferation, while the double knockout animals showed a much more pronounced reduction.
11. Please, provide a better description of how the technique shown in Figure 6A and B was performed. Please, make clear if a WT animal was used. Furthermore, statistical analysis is lacking in Figure 6B.
12. In Figure 6D, the PCR analysis was carried out using only two animals. Is it possible to perform statistical analysis with a so limited number of animals? Do the authors believe that the statistical analysis and the significant difference they found have some biological relevance? As commented before, this referee strongly recommends increasing the sample size for the minimum of five animals per analysis in each experimental group.
13. In figure S12, more gene levels were increased after the agonist's administration (IFN- α /IFN- λ). Why have the authors just evaluated the role of AREG (Amphiregulin) on DSS-colitis?
14. In the material and methods section (statistical analysis), the level of significance was not mentioned. I recommend using only the $P < 0.05$ level of significance throughout the entire MS.

Reviewer #1 (Gut immunity, mucosal barrier)(Remarks to the Author):

McElrath et al have induced colitis by DSS in IFN receptor *Ifnar1* and *Ifnlr1* knock out animals. These double knock-out animals get more severe colitis, whereas the single ones are close to WT. Most importantly, transfer of WT cells to the double knock-out limits the disease. Furthermore, amphiregulin IP injections four times, something that rescued the animals.

This manuscript contains potentially interesting results, especially the beneficial effect of amphiregulin. However, the results and conclusions does not make it up to a consistent understanding and explanation for the observations. The authors introduce and discuss the role of the epithelium, but that bone marrow transplant from WT to the double knock-out reverts the colitis phenotype elegantly rule out any role of the epithelium. The results of Fig. 5A suggests that it is the *Ifnar1* cells that are important for the hematopoietic cells to decrease colitis, but instead of following this up, the author go to the epithelium (Fig. 5D). A lot of information is lacking. How are the IFN responding cells affecting the epithelium? How is the interesting amphiregulin effects mediated and linked to cells responding to IFN?

Reviewer #1 seems to selectively focus on one set of bone marrow chimeric mice and overlooks other bone marrow chimeras. However, the fact that WT animals transferred with bone marrow cells from double interferon receptor deficient animals also survive, clearly shows the important role of IFN signaling within the epithelial compartment for amelioration of DSS-induced colitis. The authors then present data in Fig. 5D showing the compartmentalized response in the colon, with type I IFN acting on lamina propria cells and IFN-λ acting on epithelial cells. Therefore, when combined together, results obtained with bone marrow chimeric mice and observed compartmentalized IFN response in the colon, reveal the importance of type I IFN signaling in hematopoietic compartment and type III IFN signaling in the epithelial compartment, and that the intact IFN signaling in EITHER compartment is sufficient to support recovery from DSS-induced colitis. So it is unclear to us what the Reviewer #1 means stating that "bone marrow transplant ... elegantly rule out any role of the epithelium." To the contrary,

*bone marrow chimeras with intact IFN signaling ONLY in epithelial compartment (Ifnar1^{-/-}Ifnlr1^{-/-} → WT irradiated hosts; **Fig. 5A, 5B**), demonstrates that IFN signaling in epithelial compartment ALONE is sufficient to protect mice against DSS-induced colitis; and since colon epithelial cells responds only to type III IFN and not to type I IFNs (**Fig. 5D**), it is specifically type III IFN-mediated activities in epithelial cells which are sufficient to support recovery from DSS-induced colitis.*

*Reviewer #1 also comments in the above paragraph that "A lot of information is lacking. How are the IFN responding cells affecting the epithelium? How is the interesting amphiregulin effects mediated and linked to cells responding to IFN?" We believe that we provided compelling evidence that amphiregulin produced by either lamina propria cells in response to type I IFNs or epithelial cells in response to type III IFNs, supports the regeneration of epithelium following the DSS-induced injury. In the revised manuscript, we also provide additional data (**Fig. S14**), demonstrating that IFN-λ promotes migration of epithelial cells through the EGFR signaling pathway. In addition, we also included a new paragraph in the Discussion section (pages 13-14) of the revised manuscript, describing mechanisms by which AREG promotes re-epithelialization and regeneration of intestinal epithelium.*

There are number of questions (and ways of presenting the results) that need to be addressed to make this understandable. The conclusions in the last sentences (line 458) that IFN promote epithelial generation and that this is mediated by amphiregulin require further consideration and experimentation.

*As stated above, we now provide additional data (**Fig. S14**, page 10) and discussion (pages 13-14) about the actions of amphiregulin toward promoting epithelial regeneration.*

Additional MAJOR

1. The authors claim that the double KO show goblet cell depletion. In Fig. 2B mucus plugs can be observed in two crypts, suggesting what is the typical situation and misunderstanding in colitis, a faster goblet cell emptying. This has to be addressed by staining for precursor forms of typical goblet cell proteins like Mus2 and Clca1.

The authors were not aware about "the typical misunderstanding" in colitis and apparent controversy between goblet cell depletion versus "a faster goblet cell emptying". Moreover, our attempts to find relevant articles by searching PubMed for key words goblet cells and emptying" or "goblet cells and degranulation" or "goblet cells degranulation and colitis" produced no relevant results. We would appreciate if Reviewer #1 could provide a reference where we can read more about this issue to correctly address his comment. In any case, our data show the reduced number of colon epithelial cells with mucin granules, indicative of goblet cells. We did not aim to address whether this reduced staining is due to goblet cell depletion, aberrant degranulation or impaired re-generation of goblet cells. We can state in the manuscript that the reduced staining can be due to one of these reasons, if Reviewer #1 prefers.

2. The claimed data showing mucus staining is not included although mentioned in text and figure legend S3.

We apologize for the accidental omission of mucus data in Fig. S3. These data are now provided in Fig. S4 of the revised manuscript.

3. Figures lack magnification bars and seems to be shown with different magnifications.

Scale bars are now included in all images.

4. The staining for Ki-67 cells seems to suggest fewer such cells in the double KO, but the figure is of lower magnification. Even if this is the case, this does not show decreased proliferation and that this is 'responsible for the near total mortality' (line 210).

*The quantified data for Ki-67 staining are presented in Fig. 4B and show strong and highly reproducible ($P \leq 0.0001$) reduction of the number of Ki-67+ cells per crypt in DKO mice. Images of higher magnification can be included as supplementary materials. Ki-67 is well accepted as a marker of cell proliferation and staining for Ki-67 is widely used to assess proliferation of the cells in tissues. Reviewer quoted a part of a sentence that states the following: these data suggest that the combined loss of type I and type III IFN signaling results in decreased proliferation of colonic epithelial cells following DSS insult, and is responsible for the near total mortality observed in *Ifnar1^{-/-}Ifnlr1^{-/-}* mice. At this point, the accumulative data is only suggestive. However, we do believe that the combined data presented within the entire manuscript provide a compelling evidence that the lack of IFN signaling in both epithelial and hematopoietic compartments results in diminished levels of AREG expression which are insufficient to support adequate proliferation of epithelial cells. In turn, this impaired proliferation of epithelial cells leads to the compromised repair of the damaged epithelium and results in near total mortality observed in DKO mice in response to DSS treatment.*

5. The RNA Seq data claimed (line 290) is not included. Only a Fig. S10. The raw data must be presented or at least the parts that related to AREG. Any difference of the two tested IFNs. Proof that the analyzed cells were the epithelial cells? PCR of total homogenate is not a good control for expression in epithelial cells.

The raw RNAseq data is now included in the revised manuscript and there is an extended discussion of the RNAseq data related to AREG. We want to emphasize that the RNAseq data were used to identify pathways and genes which may mediate effects of IFNs on proliferation of epithelial cells. As discussed above and in the manuscript, since data with chimeric mice showed that IFN signaling in either epithelial or hematopoietic compartment was sufficient to supports the regeneration of epithelium following DSS-induced injury, we hypothesized that effects of IFNs on proliferation of epithelial cells should be indirect. The RNAseq data suggested that AREG was a likely mediator of this effect since it was induced by IFNs and EGFR-mediated signaling is known to play important role in tissue repair and regeneration. PCR analyses of AREG

expression in total colon homogenates (Fig. 6D) as well as results of in situ hybridization (Fig. 6C) demonstrate that the levels of AREG expression are strongly up-regulated in both epithelial cells and other cell types residing in lamina propria in WT mice following DSS-treatment and this up-regulation is dependent on intact IFN signaling since it is strongly diminished in DKO mice, and only DKO mice, but not single IFNR-deficient mice, succumb to DSS treatment. We did not intend to conclude that the data presented in Fig. 6C,D demonstrate IFN-induced expression of AREG ONLY in epithelial cells. If it was not clear, we now state in the revised version on page 9 that cells residing in the epithelium and the lamina propria have diminished AREG expression in DKO mice following DSS treatment. Results of experiments presented in Fig. 6E,F also demonstrate that IFNs up-regulate AREG transcription in both epithelial and hematopoietic cells.

6. Line 564. There is no information what type of AREG and supplier was used. This is a transmembrane protein, did they use the extracellular part.

The supplier for the soluble AREG (the extracellular part of this transmembrane protein) is now provided in the revised manuscript.

Reviewer #2 (Colitis, gut immune responses)(Remarks to the Author):

General comments

McElrath et al. investigated whether the combined loss of type I and type III IFN signaling might enhance the susceptibility to DSS-induced colitis. For this purpose, the authors assessed mice lacking both types of IFN receptors and reported a marked loss of goblet cells and a diminished proliferation of epithelial cells in the colon. Furthermore, McElrath et al. reported that impaired mucosal healing in double-knockout mice is driven by decreased AREG expression, which can be up-regulated via type I and type III IFN signaling. By using bone marrow chimera experiments, they also demonstrated that IFN signaling, in either the epithelial or hematopoietic compartments, is sufficient to provide protection against DSS-induced injury and to support epithelial cell proliferation in the intestinal mucosa. Collectively, data suggests a pleiotropic function for IFN signaling in colonic epithelial regeneration following DSS-induced damage. This manuscript is well written and contains novel and also potentially clinically relevant data. Experiments are generally well-planned and with some few exceptions, they seem to be well conducted.

We thank the reviewer for a brief and accurate summary of our findings and for positive overall comments on experimental designs and execution, and presentation of the data in the manuscript.

However, this referee found some critical points in this manuscript that need an extensive and careful revision by the authors, including the need to perform new experiments, before its acceptance for publication.

We appreciate the specific major comments of the reviewer, which are mostly centered on providing additional details and statistics for the experiments. We found these comments to be very relevant and appropriate, and as requested by the reviewer, we extensively and carefully revised the manuscript to address reviewer's concerns.

Major comments

One of my major concerns about the manuscript is the very condensed method's section. More recently, there is a growing number of publications, notably in high impact journals, about the lack of reproducibility of pre-clinical studies. This has led a substantial change in the guidelines for manuscript submission in most relevant international journals, including those of the Nature group. This manuscript is not in compliance with such recommendations. For instance, the authors must provide details on the animals used: species, strain, gender, age, and body weight. Please, also provide general information about the KO mice, especially regarding the genetic background, housing and husbandry conditions (the type of cage, bedding material, breeding programming, type of food, access to food and water, environmental enrichment, etc).

In the revised manuscript, we now provide a detailed description of all animal strains, and husbandry conditions.

Another very relevant point regards the sample calculation. The authors reported different numbers of animals per group. This great variation in the number of animals per group might cause serious problems and eventually bias in the statistical analysis of the data. Are the authors convinced that the small number of animals used in some experiments is sufficient to confirm that the observed statistical differences have a biological relevance?

*The authors began the study (AOM/DSS model and DSS model) with several pilot experiments. Based on these results, it became apparent that WT and single IFN receptor KO mice were not susceptible (regarding mortality) as opposed to *Ifnar1^{-/-}Ifnlr1^{-/-}* mice. These pilot experiments also showed high reproducibility of the results and low variability within each strain, and demonstrated that 5-7 animals per group were sufficient to provide statistically significant data. However, the larger numbers of *Ifnar1^{-/-}Ifnlr1^{-/-}* mice were typically used for experiments (Fig. 1 and Fig. S2) to account for their enhanced susceptibility and ensure that sufficient number of animals would survive to the experimental endpoint. We also performed additional experiments and/or analyzed additional slides and existing specimens to bring number of animal to ≥ 5 for most experiments.*

Furthermore, to avoid any bias of data interpretation, it is important to inform whether the in vivo experiments were conducted blindly (if the experimenter conducted the experiments without prior knowledge of treatments and mouse groups). Also important, were the animals assigned randomly to the various experimental groups? There are many literature examples showing that experiments carried out without randomization of

animals in the different experimental groups produce serious bias in the analysis, consequently affecting data interpretation and the main conclusions of the study. If the animals had been previously randomized before experiments, please provide the method of randomization used. These above-raised points are essential to allow the reproducibility and the robustness of preclinical scientific papers. Indeed, many scientific journals have pointed out their critical importance and have made them mandatory for publications containing in vivo experiments. For more details, please, see the following references: Landis et al ; Nature 490, 187, 2012; Glenn and Ellis, Nature, 483, 531, 2012; Peers et al., Nature Review Drug Discovery, 11, 273,2012; Collins et al., Nature, 505, 612, 2014; Allison et al., Nature, 530, 27, 2016; Nature, 542, 409, 2017; Science 355, 234, 2017.

We are well aware of the ongoing problems with the reproducibility of scientific experiments in general and specifically in the IFN- λ -related studies, and take this issue very seriously. As stated above, we observed that the results were highly reproducible and experiments were repeated at least two times. Although, experiments were not conducted blindly (cages with DDS-containing water needed to be specifically labeled according to IACUC protocol), pathology slides and other tissue specimens were analyzed blindly. Typically experiments were conducted with mice from 2-3 litters per strain, and animals of each strain were randomly selected into the DSS-treated group to normalize for the equal number of males and females, age and initial weight across the strains.

Other comments

1. Please, provide which was the initial experimental number of animals in each group. This was not described in the methods section.

The experimental number of animals in each experimental group was provided in the figure legend. This can be added into the method section, however the authors felt and still believe that it is more transparent to state the number of animals in each group in the figure legends where results of specific experiments are presented. As requested, we now provide the initial experimental number of animals for each strain for experiments where the number of animals differed substantially. As stated above, this variation was due to the fact that we used more double IFNR-deficient mice to ensure that sufficient number of animals would survive to the experimental endpoint.

2. What means the untreated DSS group? Please, provide data of animals that did not receive the DSS solution (naïve mice). This is important for purposes of comparison with the DSS-treated animals, and also to confirm the intestinal mucosa integrity and health of naïve animals.

Untreated DSS mice (naïve mice) received regular drinking water provided by the SPF barrier facility at NJMS, Rutgers University. At steady state, the weights of 6-8 week-old mice steadily increased regardless of the strain. Histological evaluation of colons from mice of different backgrounds at steady state is provided on Fig. 4A and Fig. S4 and

*shows no difference in the intestinal mucosa integrity. Although we observed thinner mucosal layer in *Ifnlr^{-/-}* and *Ifnar^{-/-}Ifnlr^{-/-}* mice.*

3. The histological score is a non-parametric data; thus, the analysis of variance cannot be applied. Please, use a non-parametric test to analyze this data (Fig. 1E and 2 D).

As suggested, histological scores were re-analyzed using a non-parametric Kurskal-Wallis test.

4. I suggest to remove the figures 1 A and 1 B from the body of the manuscript and to transfer both figures to supplementary data, as they show that association of DSS and the carcinogen azoxymethane resulted in the death of the majority of animals.

As suggested, Fig. 1A and 1B were transferred from the body of the manuscript to supplementary data (Fig. S2).

5. Figure 2D is quite confused. I suggest to remove it and to describe the corresponding data in the results section.

As suggested, Fig. 2D was removed and the data are described in the Results section.

6. In figure 4A, it appears that animals did not receive a DSS solution and in figure 4B, animals were treated with DSS. Please clarify these points in the figure legend.

As suggested, the figure legend was updated to clarify these points.

7. Authors, please, check in the figures 1A and 1C if the ANOVA test was really applied since survival curves are depicted.

The figure legends for Fig. 1 and Fig. S2 (since some data was moved to the supplementary Fig. S2 as suggested by Reviewer #2; see comment #4 above) was updated to clarify where the ANOVA test was used.

8. The authors represent the figures with Mean \pm Standard Error of the Mean (SEM), but the ARRIVE (Animal Research: Reporting of In vivo Experiments) guideline recommends the use of Standard Deviation (SD).

Authors looked at several articles that were recently published on various topics of IFN- λ biology in high impact journals including those of Nature publishing group and found that Mean \pm SEM is used more often than Mean \pm SD. For example, mean \pm SEM is used in the following papers: Galani et al., 2017, Immunity (PMID: 28514692); Broggi et al., 2017, Nat. Immunol. (PMID: 29234323); and Ye et al., 2019, Nat. Immunol. (PMID: 31201377). Therefore, we prefer to follow this common presentation style.9. In figure 4B, the representative panel for *Ifnar1* knockout mice seems to present more intense labeling than that observed for WT or *Ifnlr1* mice. However, the quantification of the data shows different results. Please check this point.

*The authors agree that this observation could be made upon the first glance. However, the crypts of WT and *Ifnlr^{-/-}* mice become elongated, as presented in the picture. This is*

indicative of rapid proliferation and is not observed in $lfnar^{-/-}$ or $lfnar^{-/-}lfnlr^{-/-}$ mice. The staining is therefore spread out over a larger crypt length and appears less intense in WT and $lfnlr^{-/-}$ mice. Thus, counting the number of positive cells per crypt, which is shown in Fig. 4B graph, revealed the reduced number of Ki-67⁺ cells per crypt in both $lfnar^{-/-}$ or $lfnar^{-/-}lfnlr^{-/-}$ mice. Crypt elongation is described in the Results section of the revised manuscript.

10. Please, explain why the animals of the $lfnar1$ group show almost an equal proliferation, while the double knockout animals showed a much more pronounced reduction.

The bone marrow chimera experiments have demonstrated that IFN signaling in either the epithelial or hematopoietic compartment supports recovery following DSS insult. The hematopoietic compartment responds to type I IFNs and levels of AREG expression are up-regulated in this compartment in response to type I IFNs. The epithelial compartment responds to type III IFNs and levels of AREG expression are up-regulated in this compartment in response to type III IFNs. The IFN-driven up-regulation of the levels of AREG expression in either compartment is sufficient to provide adequate re-generation of the intestinal epithelium following DSS-induced injury. Since type III IFN signaling is intact in the IFNAR1-deficient mice, levels of AREG expression are up-regulated in the epithelial compartment in response to type III IFNs, and this is sufficient to provide adequate protection. In the double IFNR-deficient mice, levels of AREG expression are not up-regulated by IFNs in either compartment, resulting in strongly impaired proliferation of IECs following DSS-treatment. In other words, the simultaneous loss of BOTH type I and III IFN signaling in mice, when IFN signaling in BOTH epithelial and hematopoietic compartments is affected, ONLY these DKO animals have strongly reduced epithelial proliferation following exposure to DSS.

11. Please, provide a better description of how the technique shown in Figure 6A and B was performed. Please, make clear if a WT animal was used. Furthermore, statistical analysis is lacking in Figure 6B.

The method section and the figure legend were updated to provide clarification regarding the IVIS technique. Additionally, authors also added statistical analysis for the kinetics data in Fig. 6B.

12. In Figure 6D, the PCR analysis was carried out using only two animals. Is it possible to perform statistical analysis with a so limited number of animals? Do the authors believe that the statistical analysis and the significant difference they found have some biological relevance? As commented before, this referee strongly recommends increasing the sample size for the minimum of five animals per analysis in each experimental group.

The PCR results were from one out of two independent experiments, which both generated similar results. This experiment was repeated one more time to increase the sample size to 5 animals per experimental group. Regarding biological relevance, all

the data combined strongly support our overall conclusion that the lack of IFN-driven up-regulation of the levels of AREG expression in DKO mice following DSS treatment is responsible for the increased susceptibility of DKO mice to DSS. Significant differences in the levels of AREG expression in WT and DKO mice following DSS treatment were demonstrated not only by RT-PCR in colon tissue homogenates (now re-confirmed with additional number of animals), but more importantly by in situ hybridization staining for AREG in colon tissue slides. Here, the enhanced staining is restricted to areas localized around ulcerations in WT mice, highlighting that AREG expression is increased in regions where its biological function would be beneficial for tissue repair and regeneration. It is apparent that this enhanced AREG staining is not observed in DKO mice.

13. In figure S12, more gene levels were increased after the agonist's administration (IFN- α /IFN- λ). Why have the authors just evaluated the role of AREG (Amphiregulin) on DSS-colitis?

As discussed, most genes, which expression levels were up-regulated, represent classical ISGs. Many ISGs when overexpressed have anti-proliferative and pro-apoptotic effects on cells. We were specifically looking for pathways and signals that can stimulate proliferation of the cells. EGFR signaling pathway and specifically EGFR ligand AREG have been shown to play an important role in tissue repair. Since RNAseq analysis revealed that levels of AREG expression are strongly induced by IFNs in IECs, we thought that this pathway, which has not been known to be activated by IFNs, may be involved in the increased sensitivity of DKO mice to DSS-induced colitis and decided to explore this possibility experimentally.

14. In the material and methods section (statistical analysis), the level of significance was not mentioned. I recommend using only the $P < 0.05$ level of significance throughout the entire MS.

As recommended we now state in the Materials and Methods that The P value < 0.05 indicated statistically significant differences. Specific P values are also presented for each experiment in figure legends, as it is customarily done in most publications.

Reviewers' comments:

Reviewer #2 (Remarks to the Author):

The relevance of IFNs type I and III in the healing of intestinal mucosa so far remains unexplored. In the article, the authors report that the combined loss of signalling from both type I and type III IFN receptors increases the susceptibility to inducing colitis with DSS. KO animals for both receptors exhibited a reduction in goblet cells and decreased proliferation of epithelial cells, impairing the healing of the colon epithelium. The data also shows that IFN stimulates epithelial cell migration dependent on the epithelial growth factor receptor (EGFR), leading to reepithelization. In addition, EGFR also regulates the proliferation and differentiation of stem cells in crypts. The treatment of double KO mice for both IFN receptors with AREG (EGFR ligand) restores epithelial cell proliferation, suggesting that the positive regulation of AREG is dependent on IFN. In the previous referee analysis of this MS, we recommended caution regarding its acceptance for publication, due to the needed of additional experiments and also to increase the number of animals in some experimental groups. Also, many aspects of the methodology employed by the authors needed to be better described and we suggested an extensive revision of this section.

The revised version of the manuscript is improved when compared with the previous one. Authors have performed many experiments suggested by all referees and also clarified most points raised in the referee's comments. Therefore, in my opinion the revised version of the manuscript is now suitable for publication in Nature Communications.

Reviewer #3 (Remarks to the Author):

This reviewer did not review the original version of the manuscript but has been asked to provide a review with particular emphasis on how the comments provided by original Reviewer #1 have been addressed. Generally most of the comments are addressed adequately, I call attention to two major issues:

1. Regarding the discussion of interpretation of the bone marrow chimera experiments. I accept the arguments made by the authors re the complex interpretation of the chimera experiments that in essence show that receptor signalling in both haematopoietic and epithelial compartments are important.

2. Re the discussion of goblet cell "depletion". This is not simply an academic argument as it is important to distinguish between loss of goblet cells and loss of stored granules. As goblet cell secretion increases the smaller theca will lessen the chance of sectioning and identifying alcian blue positive mucin granules and cell number can appear to decrease without doing so (staining for mucin precursor protein can address this). However, at the other extreme when compound exocytosis has occurred goblet cells are actually depleted and this occurs as DSS colitis becomes more extreme as can be seen in the micrographs. The authors offer a caveat around this that should be added to the results/discussion. The mucus measurements added to the revision help, but I am somewhat perplexed by the very small error bars presented in these data as these techniques are quite problematic technically and subject to sampling bias. The authors should carefully describe how they achieved these data and preferably show the individual data points.

Some of my own observations:

3. I have concerns about the primary DSS data shown in Fig 1B. It is claimed that the 12% of DKO

mice that survive DSS regained weight but there were only n=9 in this group so this would be one mouse. However, there are error bars on the recovery phase - please explain. An appropriately powered repeat experiment may be required. Then in Fig 1C data it is not clear whether this is a second experiment or data from mice that succumbed in the survival experiment (in which case it is biased selection of the most severe). Please clarify.

4. Line 197 attempts to tie a diminished proliferation with goblet cell loss in an argument that I think is not substantiated. The goblet cell loss is primarily due to death of mature differentiated goblet cells undergoing compound exocytosis, not due to a lack of regeneration.

5. Why is the epithelial damage so much less in the sections from day 8 after DSS in Fig 4 vs the day 7 sections in Fig 3? Were the least damaged areas selected for Ki67 analysis - please clarify.

6. Details of the AREG administration need to be added to the Fig 7 legend so the experiment is interpretable.

We again would like to thank all reviewers for the evaluation of our work and providing detailed and substantive comments. We are glad that Reviewer #2 found our revised manuscript suitable for publication in *Nature Communications*. His/her previous comments were very helpful for improving the manuscript flow and data presentation. We also found comments of Reviewer #3 very relevant and useful and believe that we adequately addressed all his/her concerns. Of note, a part of the comment #2 from Reviewer #3 seems to be missing (highlighted in yellow), so the authors addressed this comment to the best of their understanding. In addition, the reference to Fig. 3 in the comment #5 seems to be misquoted, since there are no colon sections presented in Fig. 3. The authors believe that the reference was meant for sections shown in Fig. 2, and addressed this comment as such. If our interpretations were incorrect, the authors would be happy to provide additional clarifications for these comments. Our responses to each specific comment are presented below and italicized.

Reviewer #2 (Remarks to the Author):

The revised version of the manuscript is improved when compared with the previous one. Authors have performed many experiments suggested by all referees and also clarified most points raised in the referee's comments. Therefore, in my opinion the revised version of the manuscript is now suitable for publication in *Nature Communications*.

We are happy to hear this opinion.

Reviewer #3 (Remarks to the Author):

This reviewer did not review the original version of the manuscript but has been asked to provide a review with particular emphasis on how the comments provided by original Reviewer #1 have been addressed. Generally most of the comments are addressed adequately, I call attention to two major issues:

1. Regarding the discussion of interpretation of the bone marrow chimera experiments. I accept the arguments made by the authors re the complex interpretation of the chimera experiments that in essence show that receptor signalling in both haematopoietic and epithelial compartments are important.

The authors are glad that our arguments and data interpretations were found acceptable and convincing by Reviewer #3.

2. Re the discussion of goblet cell "depletion". This is not simply an academic argument as it is important to distinguish between loss of goblet cells and loss of stored granules. As goblet cell secretion increases the smaller theca will lessen the chance of sectioning and identifying alcian blue positive mucin granules and cell number can appear to decrease without doing so (staining for mucin precursor protein can address this). However, at the other extreme when compound

exocytosis has occurred goblet cells are actually depleted and this occurs as DSS colitis becomes more extreme as can be seen in the micrographs. The authors offer a caveat around this that should be added to the results/discussion. The mucus measurements added to the revision help, but I am somewhat perplexed by the very small error bars presented in these data as these techniques are quite problematic technically and subject to sampling bias. The authors should carefully describe how they achieved these data and preferably show the individual data points.

As suggested by Reviewer #3, sections related to goblet cell staining were revised to indicate that the observed reduced staining for mucin granules could be due to goblet cell depletion, aberrant degranulation or impaired re-generation of goblet cells.

We have also provided an extended description for the "Analysis of inner mucus layer" in the Methods and would like to point out that representative images of inner mucus layers in the colon from different mouse strains are presented in Fig. S4C and the individual data points are presented in the Source data file.

Some of my own observations:

3. I have concerns about the primary DSS data shown in Fig 1B. It is claimed that the 12% of DKO mice that survive DSS regained weight but there were only n=9 in this group so this would be one mouse. However, there are error bars on the recovery phase - please explain. An appropriately powered repeat experiment may be required. Then in Fig 1C data it is not clear whether this is a second experiment or data from mice that succumbed in the survival experiment (in which case it is biased selection of the most severe). Please clarify.

Number of DKO mice used to obtain the experimental data shown in Fig. 1B was n=17, not n=9; n=9 refers to IFNLR single KO mice (number of animals of each strain follows, not precedes the strain abbreviation in the Fig. 1 legend: WT, n=11; Ifnar1^{-/-}, n=6; Ifnlr1^{-/-}, n=9; Ifnar1^{-/-}Ifnlr1^{-/-}, n=17). Two DKO mice survived in this experiment (12%; 2 out of 17 mice) and recovery curve shows average weight with error bars of two survived mice. As stated in the figure legend, data shown in Fig. 2A, 2B are representative of two independent experiments. For the repetition experiment, the number of animals was as follows: WT, n=5; Ifnar1^{-/-}, n=7; Ifnlr1^{-/-}, n=6; Ifnar1^{-/-}Ifnlr1^{-/-}, n=16. Two DKO mice out of 16 mice survived in the repetition experiment, demonstrating high reproducibility of the results. Of note, similar percent survival was observed in experiments with the AOM/DSS treatment protocol where experiments were repeated more than twice to obtain number of DKO mice, which survived the treatment (Fig. S2A), sufficient for tumor burden analyses (Fig. S2B). In all these experiments we consistently observed that ~10% of mice survive DSS treatment with or without AOM treatment. Therefore, we are highly confident in the reproducibility of the data presented in Fig. 1B.

Data shown in Fig. 1C, 1D and 1E and Fig. S3 are derived from independent experiments (repeated twice) where all mice were sacrificed at day 11. This point is now clarified in the Fig. 1 legend.

4. Line 197 attempts to tie a diminished proliferation with goblet cell loss in an argument that I think is not substantiated. The goblet cell loss is primarily due to death of mature differentiated goblet cells undergoing compound exocytosis, not due to a lack of regeneration.

The sentence on line 197 has been revised following reviewer's comments.

5. Why is the epithelial damage so much less in the sections from day 8 after DSS in Fig 4 vs the day 7 sections in Fig 3? Were the least damaged areas selected for Ki67 analysis - please clarify.

We are unclear about reviewer's comment #5. Our best guess is that the reviewer compares images shown in Fig. 4B that were used to assess Ki-67 staining with images presented in Fig. 2A and 2B (not Fig. 3) showing H&E and PAS/Alcian blue stainings. If that is indeed the case, we would like to point out that stainings in Fig. 2 are shown for mice sacrificed on day 11 (not day 7) after the start of DSS treatment. H&E and PAS/Alcian blue stainings for the day 8 of the DSS protocol are presented in Fig. S5 and demonstrate epithelial damage similar to that shown in Fig. 4). As discussed in the manuscript, the reduced Ki-67 staining and therefore the impaired proliferation of epithelial cells is already apparent at day 8 of the DSS protocol in the DKO mice colon, while the degree of colon tissue damage is still comparable amongst all mouse strains. If reviewer's comment was meant for another set of images, we would appreciate further clarification from the reviewer.

6. Details of the AREG administration need to be added to the Fig 7 legend so the experiment is interpretable.

Details of the AREG administration were added to the Fig. 7 legend, as suggested.